# Most primary olfactory neurons have individually neutral effects on behavior

**Tayfun Tumkaya[1,2], Safwan Burhanudin[1], Asghar Khalilnezhad[1], James Stewart[1], Hyungwon Choi[1,3], Adam Claridge-Chang[1,2,4]***

[1]Institute for Molecular and Cell Biology, A*STAR, Singapore, Singapore; [2]Program in Neuroscience and Behavioral Disorders, Duke NUS Graduate Medical School, Singapore, Singapore; [3]Department of Medicine, National University of Singapore, Singapore, Singapore; [4]Department of Physiology, National University of Singapore, Singapore, Singapore

**Abstract** Animals use olfactory receptors to navigate mates, food, and danger. However, for complex olfactory systems, it is unknown what proportion of primary olfactory sensory neurons can individually drive avoidance or attraction. Similarly, the rules that govern behavioral responses to receptor combinations are unclear. We used optogenetic analysis in *Drosophila* to map the behavior elicited by olfactory-receptor neuron (ORN) classes: just one-fifth of ORN-types drove either avoidance or attraction. Although wind and hunger are closely linked to olfaction, neither had much effect on single-class responses. Several pooling rules have been invoked to explain how ORN types combine their behavioral influences; we activated two-way combinations and compared patterns of single- and double-ORN responses: these comparisons were inconsistent with simple pooling. We infer that the majority of primary olfactory sensory neurons have neutral behavioral effects individually, but participate in broad, odor-elicited ensembles with potent behavioral effects arising from complex interactions.

*For correspondence:
claridge-chang.adam@duke-nus.edu.sg

Competing interest: The authors declare that no competing interests exist.

## Editor's evaluation

Olfactory coding is still an open question in neuroscience. Therefore, this paper is of potential interest to a broad audience of neuroscientists. It undertakes a thorough investigation of how olfactory sensory neurons drive avoidance or attraction in flies and also addresses how combinations of active ORNs can become behaviorally meaningful. It has great potential value for clarifying how animals map sensory input to valence.

## Introduction

Animals interact with their environment using motor functions that are guided by information that enters the brain from multiple sensory systems. These diverse sensory inputs are thought to interact with each other, with previously stored information, and with the internal physiological state of the animal to elicit a more or less appropriate behavioral response. Two central problems of neuroscientific research are (1) how individual sensations influence behavior and (2) how multiple streams of sensory information are reconciled into meaningful behavior. The *Drosophila* olfactory system is an effective model to address these critical questions (*Couto et al., 2005*; *Eisthen, 2002*; *Wang et al., 2003*), facilitated by powerful genetic approaches, the ability to handle large sample sizes, and the numerical simplicity of *Drosophila* neural systems. Flies detect odors with their antennae and maxillary palps, which together contain ~1300 olfactory-receptor neurons (ORNs) (*Lai et al., 2008*). The odor-response profile of each adult ORN is determined by one of ~45 possible receptor types (*Fishilevich*

and Vosshall, 2005; Gomez-Diaz et al., 2018). ORNs sharing the same receptor type converge on a glomerulus in the antennal lobe, where they synapse with local interneurons (LNs) and projection neurons (PNs) (Couto et al., 2005; Gao et al., 2000). Innervating throughout the antennal lobe and connecting multiple glomeruli, the LNs facilitate both excitatory and inhibitory interactions between glomeruli (Groschner and Miesenböck, 2019). This modified information is relayed by the PNs to higher brain centers, namely mushroom bodies and the lateral horn (Lai et al., 2008; Wang et al., 2014; Wong et al., 2002). The distinct nature of the ORN types allows us to consider each type as a single channel of information input. Mapping how ORNs steer behavior would inform a broader understanding of how sensory circuits influence behavioral output.

Odor-induced activity in ORNs can trigger approach and avoidance behaviors, collectively referred to as behavioral 'valence' (Knaden et al., 2012). At least some ORN-driven behaviors appear to follow simple rules: a subset of receptors respond specifically to single odorants, and their ORNs individually drive innate valence (Ache and Young, 2005; Grabe and Sachse, 2018; Haddad et al., 2010; Haverkamp et al., 2018; Stensmyr et al., 2012; Suh et al., 2007). These acutely tuned, strongly valent ORN classes include neurons tuned to danger (e.g. toxic odorants) and pheromones. Given the direct relationship between such odors and valence, these ORN types and their associated downstream pathways have been termed 'labeled lines' (Grabe and Sachse, 2018; Hildebrand and Shepherd, 1997; Kurtovic et al., 2007). The existence of labeled lines proves that at least some olfactory behaviors follow simple ORN-activity rules.

Unlike labeled lines, many other olfactory receptors are broadly tuned to respond to many odorants, and most pure odorants evoke responses across many ORN classes (Hallem and Carlson, 2006). As most olfactory behavior relies on activity in ORN groups, there is the outstanding question of how individual channels contribute to an odor's overall valence. It is not known how much more complex multi-ORN valence is compared to the relative simplicity of labeled-line behavior. Earlier studies have looked at whether multi-glomerular olfactory valence could be explained by statistical models of ORN or PN activity patterns. Depending on the type of experiments, some found that valence could be explained by simple rules, for example weighted summation of larval ORN activity (Kreher et al., 2008). Other studies found no relationship between single-glomerulus properties and odor-evoked behavior, or invoked more complex models of antennal-lobe function (Badel et al., 2016; Knaden et al., 2012; Kuebler et al., 2012; Meyer and Galizia, 2011). Due to the many–many relationship of most odorants and ORNs, using natural odors to isolate single-ORN valence effects is challenging (Haddad et al., 2010; Knaden et al., 2012; Semmelhack and Wang, 2009; Thoma et al., 2014; Turner and Ray, 2009). One study overcame this challenge by activating single-ORN types optogenetically (Bell and Wilson, 2016). Using eight attractive ORN types the researchers found that two-way ORN valence combinations follow either summation or max-pooling; this supports the idea that olfactory valence arises from simple rules. Thus, both simple mechanisms (labeled lines, summation) and complex inter-channel interactions have been invoked to explain olfactory valence, but their relative importance remains controversial.

The present study had two primary aims: to map which single ORN types drive valence; and to examine the extent to which simple pooling rules govern ORN–valence combinations. To do so, we measured the valence coding of the primary olfactory system by optogenetically stimulating single ORN classes. In the wild, olfaction typically occurs in windy environments, and is influenced by hunger state, so we explored whether single-type-ORN valence is similarly contingent on these factors (Bell and Wilson, 2016; Sengupta, 2013). We activated pairs of ORN types to investigate how their combinations influence valence, and built statistical models of ORN interactions. All the results indicate that ORN–valence computations are complex.

## Results

### An optogenetic behavior assay reports on sensory valence

We investigated which single classes of primary chemosensory neurons can elicit preference behavior, and the capability of the present experimental system to replicate prior studies. Generating *Drosophila* lines that express the red-light-shifted channelrhodopsin CsChrimson (Chr) in different receptor neurons, we tested whether individual neuronal types drive attraction or avoidance. Flies were presented with a choice between light and dark environments in a wind- and light-induced

self-administration response (WALISAR) apparatus (*Figure 1A*). We tested the validity of this approach with flies expressing the Chr channel under the control of the *Gr66a-Gal4* driver line, which labels bitter-taste-sensing gustatory-receptor neurons (*Moon et al., 2006*) and has been previously reported to drive avoidance when artificially activated (*Aso et al., 2014*; *Shao et al., 2017*). In a sequence of 12 trials using three light intensities (14, 42, and 70 µW/mm$^2$) and two airflow states (on/off), the experimental *Gr66a > Chr* flies displayed robust light avoidance (*Figure 1B–F*; *Figure 1—figure supplement 1B–1D*; *Figure 1—figure supplement 2*). To benchmark the assay against a published method, we compared present data with prior results (*Shao et al., 2017*); the standardized effect size of Gr66a-neuron avoidance at the highest intensity (Cohen's $d = -2.70$ at 70 µW/mm$^2$) was almost as large as that of the previously reported valence response ($d = -3.63$; *Figure 1—figure supplement 3A*). This replication of aversive Gr66a-cell activation confirmed that this study's optogenetic-choice apparatus could be used to measure negative valence mediated by sensory neurons.

To benchmark attraction behavior, we used an *Orco-Gal4* driver line that labels ~70% of all ORNs (*Larsson et al., 2004*; *Wang et al., 2003*). Others have reported *Orco*-neuron valence results: one study showed no behavioral effect *Suh et al., 2007*; another found attraction in the presence of a wind cue only *Bell and Wilson, 2016*. In our experiments, at two higher light intensities (42 and 70 µW/ mm$^2$), *Orco > Chr* flies exhibited pronounced attraction even in still air (*Figure 2G*; *Figure 2—figure supplement 1*). The valence was typically stronger than that reported in prior studies (e.g. $d = 0.56$ in still air in this study *versus* $d \leq 0.10$), establishing assay sensitivity for attraction (*Figure 1—figure supplement 3B*). Changing the temporal sequence of the 12 trials had negligible effects on *Orco*-neuron positive valence, suggesting that valence is not greatly susceptible to order bias, for example, due to habituation (*Figure 2—figure supplement 1*). Driver and responder controls typically had very similar distributions (*Figure 2—figure supplement 2*)**.** Together with the *Gr66a* + results, these data indicate that WALISAR is a valid, sensitive assay for measuring the valence of chemosensory circuits.

## Only one-fifth of ORN types drive valence

We aimed to estimate what proportion of single ORN types elicit valence when activated alone. Using available Gal4 lines, each driving expression in a single ORN type, we assessed the optogenetic valence of 46 receptor classes (*Figure 2A*). To separate the valence effects from noise, we analyzed data from ~7176 flies with the empirical Bayes method (*Figure 2B–C*). The empirical Bayes analysis identified 10 valent classes: six ORNs elicited attraction and four elicited aversion (*Figure 2A*). The hits included six ORN classes with identified ligands. Four are considered labeled lines: Or56a, the receptor for the aversive odorant geosmin; Gr21a, the receptor for the aversive odorant carbon dioxide; Or67d, the receptor for the pheromone 11-cis-vaccenyl acetate; and Or47b, which senses the pheromone palmitoleic acid (*Davis, 2007*; *Jones et al., 2007*; *Kurtovic et al., 2007*; *Lin et al., 2016*; *Stensmyr et al., 2012*; *Suh et al., 2004*; *van der Goes van Naters and Carlson, 2007*). Additionally, Or83c mediates attraction to farnesol, an odorant produced by some ripe fruits (*Ronderos et al., 2014*), while Or42b mediates attraction to vinegar (*Semmelhack and Wang, 2009*). Given that this screen successfully recaptured six ORN types already known to be involved in ecologically relevant valence functions, we consider that the screen was valid and sensitive. Furthermore, the majority of the hits being ORNs with already-established valence implies that most ORNs are not singly valent.

To contextualize the screen's outcome, we conducted a literature review, tabulating prior and current results in ORN valence (*Supplementary file 1*), from 16 studies including this one (*Bell and Wilson, 2016*; *Chin et al., 2018*; *Dweck et al., 2013*; *Faucher et al., 2006*; *Gao et al., 2015*; *Hernandez-Nunez et al., 2015*; *Jung et al., 2015*; *Knaden et al., 2012*; *Mathew et al., 2013*; *Poon et al., 2010*; *Ronderos et al., 2014*; *Semmelhack and Wang, 2009*; *Stensmyr et al., 2012*; *Suh et al., 2007*; *Suh et al., 2004*). Although methodological diversity precludes a formal, quantitative meta-analysis (*Borenstein et al., 2009*; *Tumkaya et al., 2018*), it is clear that—for many ORNs—a consensus on valence is lacking. For example, the ORN screen showed that two additional, pheromone-responding ORNs (Or88a and Or65a) were not Empirical-Bayes hits (*Chin et al., 2018*; *van der Goes van Naters and Carlson, 2007*); however, no prior single-ORN data have shown these ORN types to individually drive valence (*Supplementary file 1*). It is possible that the behavioral effects of Or88a and Or65a depend on the presence of other cues or the activation of other receptors.

We only used male flies in the screen. Because odor responses in female flies might differ—especially for pheromonal receptors—we checked for possible sex differences in five receptor classes:

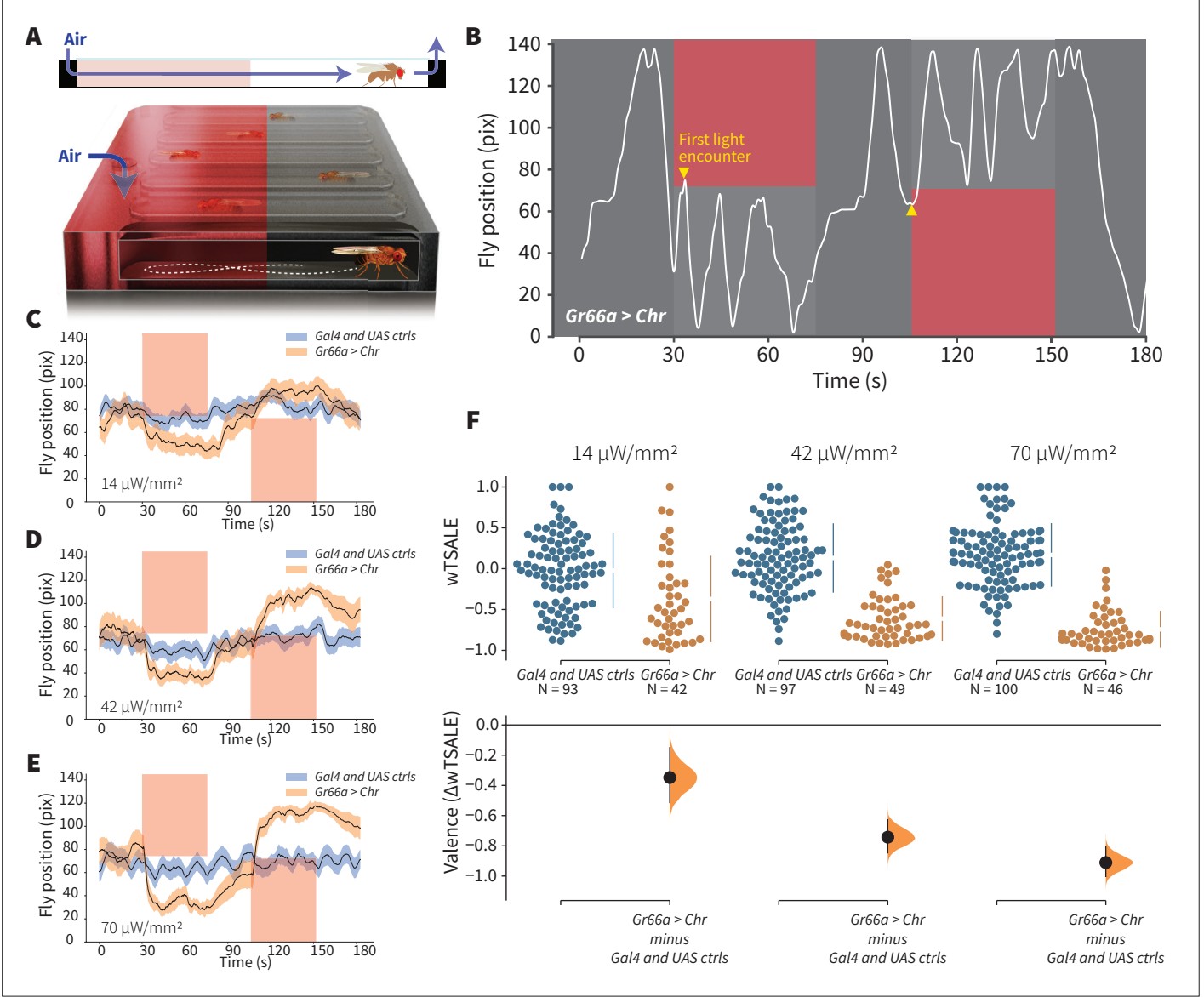

**Figure 1.** An optogenetic behavior assay reports on sensory valence. (**A**) Side view schematic (top) showing air flow path and overview (bottom) of the WALISAR assay. Individual flies were placed in chambers and given a choice between no light or red-light illumination. Air flow was used in some experiments. (**B**) A representative path of a fly displaying strong light avoidance. The white line represents the location of a *Gr66a-Gal4> UAS-CsChrimson* fly in the chamber throughout an experiment. The light preference of each fly was calculated by how much time it spent in the illuminated zones after the initial encounter with light (yellow arrows). (**C-E**) Trace plots representing the mean location of controls (*w*[1118]; *Gr66a-Gal4* and *w*[1118]; *UAS-CsChrimson*), and test flies (*Gr66a-Gal4 > UAS-CsChrimson*) throughout an experiment at 14, 42, and 70 µW/mm² light intensities. The blue and orange ribbons indicate 95% CIs for the control and test flies, respectively. (**F**) An estimation plot presents the individual preference (upper axes) and valence (lower) of flies with activatable *Gr66a+* neurons in the WALISAR assay. In the upper panel, each dot indicates a preference score (wTSALE) for an individual fly: *w*[1118]; *Gr66a-Gal4* and *w*[1118]; *UAS-CsChrimson* flies are colored blue; and *Gr66a-Gal4 > UAS-CsChrimson* flies are in orange. The mean and 95% CIs associated with each group are shown by the adjacent broken line. In the bottom panel, the black dots indicate the mean difference (ΔwTSALE) between the relevant two groups: the valence effect size. The black whiskers span the 95% CIs, and the orange curve represents the distribution of the mean difference.

The online version of this article includes the following figure supplement(s) for figure 1:

**Figure supplement 1.** An optogenetic behavior assay reports on Gr66a-induced avoidance.

**Figure supplement 2.** Occupancy analysis of the WALISAR chambers during Gr66a-neuron activation.

**Figure supplement 3.** The effect sizes in WALISAR and previously reported optogenetic valence assays are comparable.

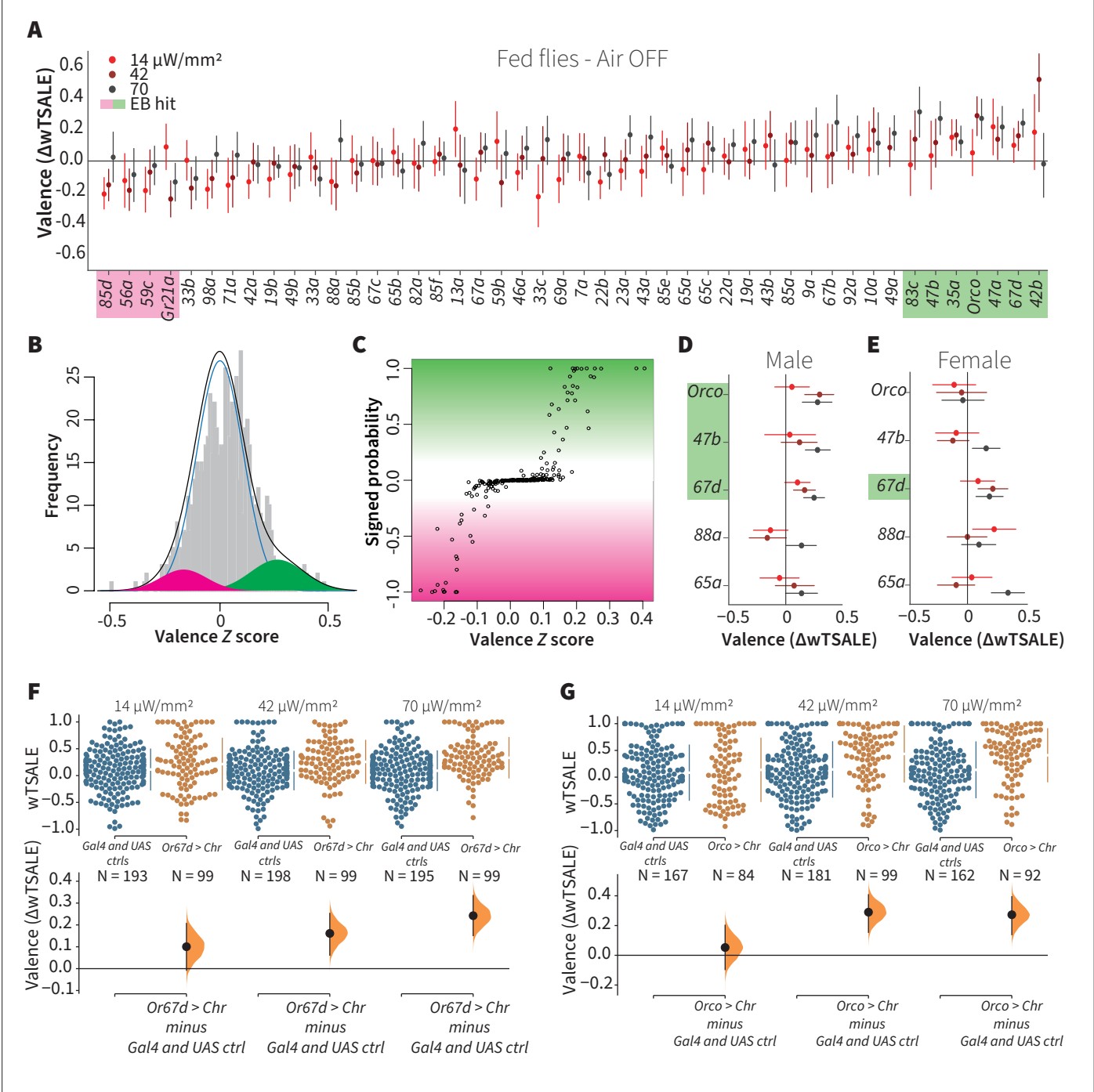

**Figure 2.** A small minority of ORN classes individually drive valence. (**A**) An effect-size plot of the valence screen of 45 single-ORN types (and *Orco* neurons). Each dot represents a mean wTSALE difference of control (N ≅ 104) and test (N ≅ 52) flies, whisker indicate 95% CIs. The shades of red represent the three light intensities. Valent ORNs are shaded with magenta (aversion) or green (attraction). (**B**) The histogram (gray bars) of the median ORN ΔwTSALE ratios, and the mixture model that is fit to the data by empirical Bayes method. The blue line represents the null distribution, while the magenta and green curves represent ORNs with negative and positive valences, respectively. The black line represents the overall distribution of the effect sizes. (**C**) The signed posterior probability that each valence score is a true behavioral change, plotted against the respective effect size (median effect across the three light intensities). (**D-E**) The olfactory valence of the *Orco* and four single-ORNs were tested in female flies (the male fly data is replotted from Panel A for comparison). The valence responses are represented as the mean difference (ΔwTSALE) of control (N ≅ 104) and test (N ≅ 52) flies, along with 95% CIs. Color key is the same as above. (**F-G**) Estimation plots show the optogenetic preference (upper panel) and valence (lower panel) of flies with activatable *Or67d* and *Orco* neurons across three light intensities. The dots in the top panels represent single flies, while the

*Figure 2 continued on next page*

*Figure 2 continued*

broken lines indicate the mean and 95% CIs. The differences between the pairs of test and control groups are displayed in the bottom panel, where the whiskers are 95% CIs and the curves are the distributions of the mean difference.

The online version of this article includes the following figure supplement(s) for figure 2:

**Figure supplement 1.** Orco-neuron activation triggers attraction regardless of experimental order.

**Figure supplement 2.** Female-fly responses to activation of Orco and four pheromone-related cells.

*Orco* cells and four pheromone-responsive ORNs (*Figure 2D–E*). Although male flies exhibited strong responses to *Orco*-neuron activation, females showed no response. Surprisingly, this lack of response in females turned into a strong attraction when they were starved (*Figure 2—figure supplement 2*). Only one pheromone-receptor class showed sexual dimorphism: activation of Or47b neurons (sensors of an aphrodisiac pheromone) was attractive to males, while females were indifferent.

Together, these results indicate that in isolation, most ORN classes do not drive valence. The presence of six known valent ORN types in the 10 hits, and the predominance of neutral ORNs suggest that most olfactory channels influence behavior only when activated in concert as part of an odor-evoked ensemble. It should be noted, however, that this initial analysis was performed on flies measured in still air—an abnormal condition in the wild.

## Wind does not amplify single-ORN valences

It has been previously reported that wind is essential for the optogenetic valence of *Orco* neurons (*Bell and Wilson, 2016*). We thus aimed to test the hypothesis that wind amplifies ORN valence, possibly eliciting valence in some otherwise non-valent ORN classes. We tested this in the same flies by also measuring the valences of 46 ORN types in the presence of airflow (*Figure 3A*). From each fly, we used the paired wind–no-wind responses to calculate wind-specific effect sizes, $\Delta\Delta$, for each light intensity and each ORN type (*Figure 3B*). With a lone exception (wind rendered Or59b valence more aversive in the lowest light intensity only), an empirical-Bayes model found that the wind effect sizes were indistinguishable from noise (*Figure 3C–D*). Contradicting our hypothesis, this result indicates that wind has essentially no impact on single ORN type-elicited behavior in walking flies. This result also generalizes the finding that, in either still or windy conditions, only a minority of ORN classes individually drive valence.

## Hunger has a limited effect on single-ORN valence

Chemosensory behaviors—for example gustatory and olfactory responses—are influenced by an animal's internal energy state. Low internal energy, for example, can sensitize food odor-responsive ORNs and drive foraging (*LeDue et al., 2016*; *Sengupta, 2013*; *Zhou et al., 2010*). We hypothesized that starvation would thus increase the single-ORN valence behavior, especially for attractive ORN types. To test this hypothesis, we assayed ~7176 starved flies for their optogenetic ORN valence (*Figure 4A*), and compared their behavior with that of the fed flies described above. Surprisingly, starvation did not enhance attraction. On the contrary, it reduced the responses triggered by three otherwise attractive ORNs: Or42b, Or47a, and Or83c (*Figure 4B*). Starvation also shifted the otherwise neutral valences of Or85f and Or49a, two receptors involved in sensing wasp odors (*Ebrahim et al., 2015*), into aversion (*Figure 4B*). Thus, hunger reduced the positive valence of a few pheromone- and food-sensing-ORNs, while slightly increasing the aversiveness of predator-sensing ORN classes. Overall, hunger does not have a broadly amplifying effect on single-ORN valence.

## ORN-valence combinations follow complex rules

Because the ORN screens showed that most single ORN types do not drive valence individually, it would appear that activity across multiple sensory channels simultaneously is required to drive most (non-labeled-line) olfactory behavior. One hypothesis of combinatorial odor valence holds that ORN-combination behaviors arise from simple two-way pooling rules: summation and max-pooling (both widely used in neural-net construction) (*Bell and Wilson, 2016*; *Goodfellow et al., 2016*).

To address this hypothesis, we asked how single-ORN valences are combined when two ORN classes are activated. We crossed eight ORN driver lines (three positive, two negative, and three neutral) so as to generate seven, two-way ORN combinations (henceforth 'ORN-combos'). Compared

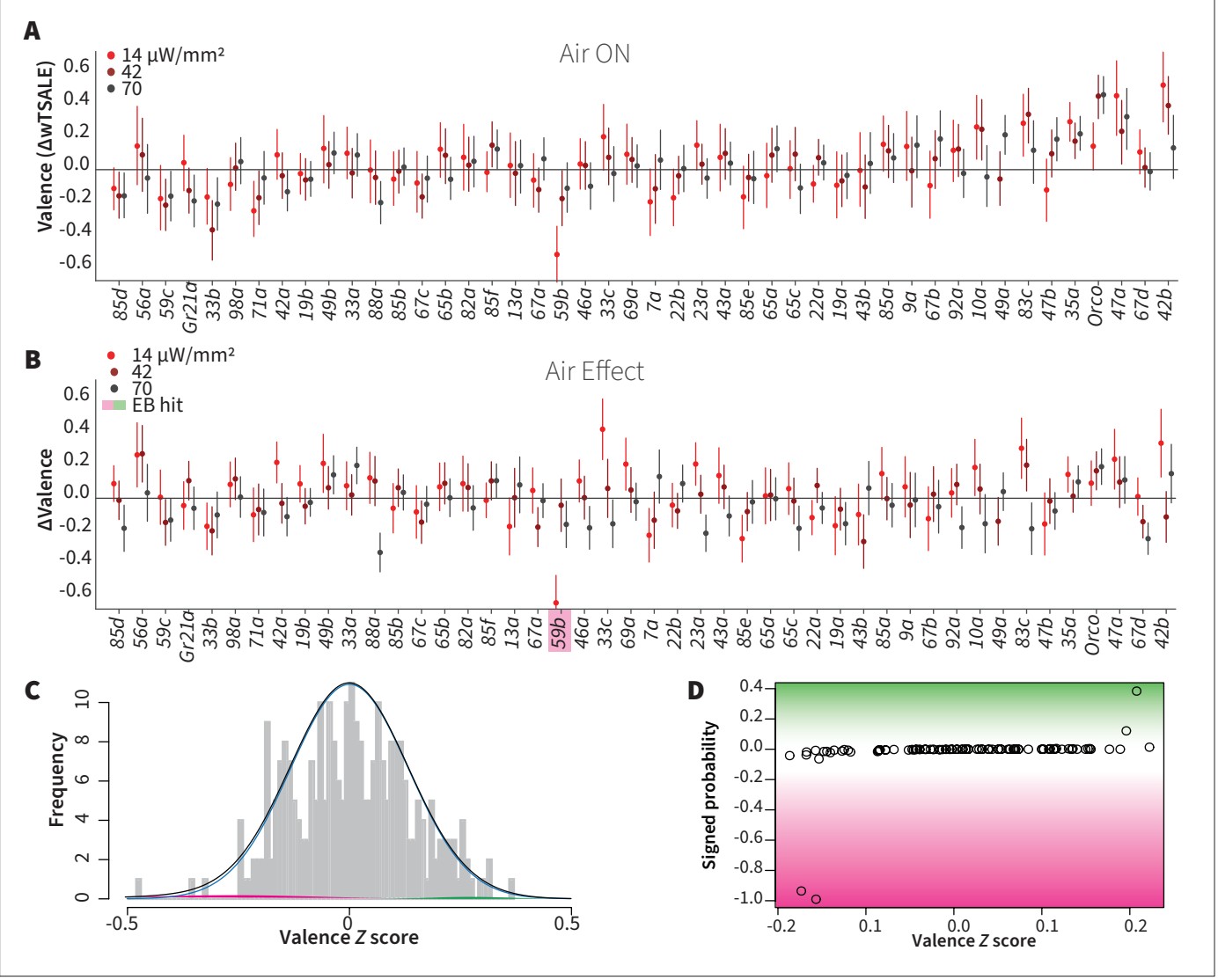

**Figure 3.** Wind does not amplify single-ORN valences. (**A**) The results of ORN valence assays in the presence of airflow. The red dots represent the mean wTSALE differences of control (N ≅ 104) and test (N ≅ 52) flies with the 95% CIs. (**B**) The differences between the effect sizes of air-on and air-off experiments (ΔΔwTSALE). The magenta shaded box indicates the sole empirical Bayes hit, Or59b, that showed an increase in aversion at the lowest light intensity only. (**C**) A statistical mixture model was fitted to the ΔΔwTSALE scores. The grey bars are the response histogram. The magenta and green curves represent the effect sizes that differ from the null distribution (blue line). The black line represents the overall distribution of the ΔΔwTSALE scores. (**D**) The signed posterior probabilities of the ΔΔwTSALE scores being true behavioral changes versus their median effect sizes. The overall probability of true wind effects was nearly zero [$p$(ΔΔwTSALE) ≅ 0.0].

to their constituent single ORNs, the ORN-combos elicited distinct valences (*Figure 5A, B and F*). We modeled the combination effect sizes with three pooling functions: summation, max-pooling, and min-pooling (*Figure 5C–E*). Strikingly, regression showed that none of the three functions could account for a large proportion of ORN-combo valence: summation, min-pooling, and max-pooling all had coefficients of determination of ~0.2 or lower (*Figure 5—figure supplement 1A-C*). Furthermore, Bland-Altman method-comparison plots revealed wide limits of agreement (LoAs) between the observed and predicted ORN-combo valences by all three models: summation [SD1.96–0.45, 0.28], max-pooling [SD1.96–0.34, 0.13], min-pooling [SD1.96–0.1, 0.35] (*Figure 5—figure supplement 1D, E, F*). This analysis thus demonstrates that none of these three simple pooling rules are major predictors of how two-ORN odor valence emerges from single-ORN valence.

To generalize this analysis, we built multiple-linear regression models of ORN-combo associations (*Figure 5—figure supplement 2*). In these models, as the effect sizes are standardized on both

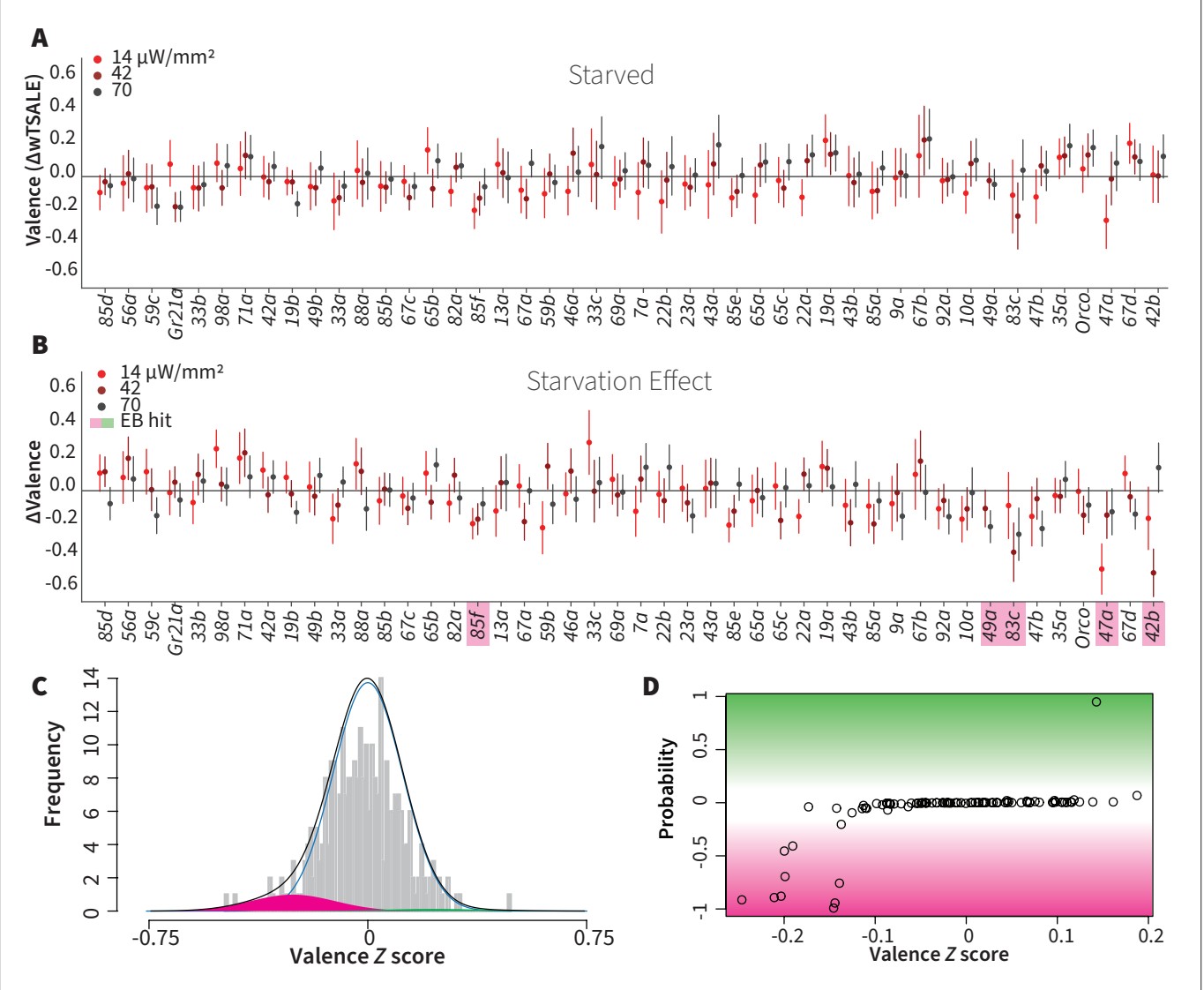

**Figure 4.** Starvation affects valence responses for five ORN classes. (**A**) An ORN valence screen for starved animals. The red dots indicate the mean wTSALE differences between control (N ≅ 104) and test (N ≅ 52) flies. (**B**) The mean differences between fed and starved flies across ORNs. The magenta shading indicates the ORNs that are affected by starvation according to the Empirical-Bayes analysis. (**C**) A histogram of the median ΔΔwTSALE ratio distribution. The blue line represents the null distribution, the magenta and green curves represent the distributions of negative and positive valences that are separated from the null distribution, and the black line represents the overall ΔΔwTSALE distribution. (**D**) The signed posterior probability of the ΔΔwTSALE scores being true behavioral changes are plotted against their median ratios.

individual ORNs and combos, the estimated β weights indicate the relative contribution strength of each of the two ORN classes. We drew scatterplots of the medians of β values for the three light intensities (*Figure 5GHI*). If the combination valences arose from summation, we would expect the β points to cluster along the diagonal (equal contribution); if combination valences followed max- or min-pooling, we would expect points clustering along the axes. However, the β points were dispersed, indicating a diversity of pooling rules. Moreover, as the light intensity increased, the β points shifted further away from the diagonal (*Figure 5J*) and, in some cases, flipped dominance (*Figure 5K*). For these ORN-type pairs, increasing intensity is associated with two phenomena: one of the two ORNs tends to become more dominant; and the combination rules are different at different activity levels. Thus, the interactions between ORN pairs vary depending on receptor identity and stimulus intensity.

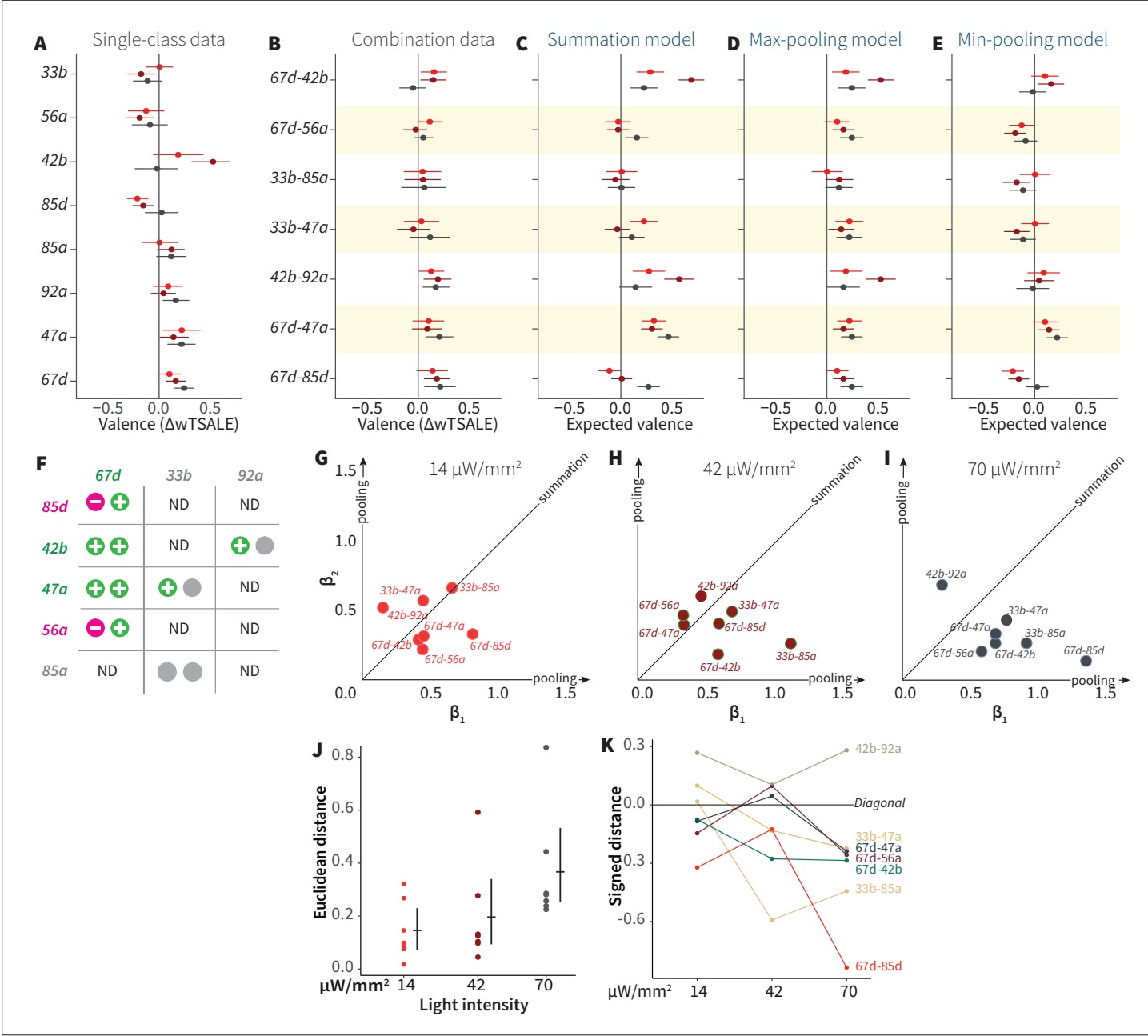

**Figure 5.** ORN-valence combinations follow complex rules. (**A**) Valence responses of the single-ORN lines used to generate ORN-combos (replotted from *Figure 1*). The dots represent the mean valence between control (N≅ 104) and test (N ≅ 52) flies (ΔwTSALE with 95% CIs). The shades of red signify the three light intensities. (**B**) The valence responses produced by the ORN-combos in the WALISAR assay in three light intensities. (**C–E**) ORN-combo valences as predicted by the summation (**C**), max pooling (**D**) and min pooling (**E**) models. (**F**) Three positive (green), two negative (magenta), and three neutral (gray) ORNs were used to generate seven ORN two-way combinations. (**G–I**) Scatter plots representing the influence of individual ORNs on the respective ORN-combo valence. The red (**G**), maroon (**H**), and black (**I**) dots indicate ORN-combos at 14, 42, and 70 µW/mm² light intensities, respectively. The horizontal ($\beta_1$) and vertical ($\beta_2$) axes show the median weights of ORN components in the resulting combination valence. (**J**) Euclidean distances of the ORN-combo β points from the diagonal (summation) line in panel J. The average distance increases as the light stimulus intensifies: 0.14 [95CI 0.06, 0.23], 0.20 [95CI 0.06, 0.34], and 0.37 [95CI 0.20, 0.53], respectively. (**K**) The β weights of the ORN-combos from the multiple linear regression are drawn as the signed distances of each ORN-combo from the diagonal line over three light intensities. The ORN weightings change magnitude and, in a few cases, the dominant partner changes with increasing optogenetic stimulus.

The online version of this article includes the following figure supplement(s) for figure 5:

**Figure supplement 1.** Linear analyses of combination valence results.

**Figure supplement 2.** Bootstrapped distributions of β weights of the ORN-combo constituents.

*Figure 5 continued on next page*

*Figure 5 continued*

**Figure supplement 3.** A linear model accounts for only 23% of the variance in odor behavior.

**Figure supplement 4.** Non-linear models suffer from the small size of the odor-valence data set.

## Single ORN information predicts odor behavior poorly

The variable interactions between pairs of ORN types support the idea that olfactory valence is determined by complex dynamics in multiple layers of downstream circuits. So we anticipated that models using ORN activity would not be strongly predictive of odor valence. To explore this, we drew on data from two previous studies that used a panel of 110 odorants: one used the odor panel to make physiological recordings from 24 ORN types—of which 23 were tested for behavioral valence in this study (*Hallem and Carlson, 2006*); the other study measured behavioral valence for all 110 odors (*Knaden et al., 2012*). We adopted a partial-least-squares discriminant analysis (PLS-DA), to reduce the 23-dimensional feature space into fewer latent variables (LVs) with internal correlation (*Figure 5—figure supplement 3A*). Various models based on various numbers of LVs could partially predict odor preference. A multiple linear regression (MLR) model with eight LVs had the best performance, with an adjusted coefficient of determination (R2adj) of 0.23. While the two-dimensional space defined by the first two LVs supports a partial separation of aversive and attractive odors (*Figure 5—figure supplement 3B*), the low R2adj score indicates that linear combinations of the available ORN-activity patterns are only weakly predictive of valence. The poor predictiveness of both the MLR model and a support-vector regression model were further confirmed by cross-validation (*Supplementary file 2* - Sheet 1). For non-linear models, performance was limited by the relatively small sample size: two models with non-linear kernels (polynomial and radial basis function) had high error rates and learning curves indicating overfit (*Supplementary file 2* - Sheet 1, *Figure 5—figure supplement 4*). Lastly, we asked whether the models could be improved with incorporation of the optogenetic valence data. Although valence-weighting improved the $R^2_{adj}$ of the MLR model from 0.23 to 0.33, the predictive performances of the models were not improved (*Supplementary file 2* - Sheet 2). This further supports the hypothesis that linear combinations of ORN activity can only account for a minority of odor preference.

## Discussion

### Most ORN classes have no individual effect on valence

This study's primary goal was to identify the proportion of ORNs that can drive valence behavior individually. The screen results indicate that 10/45 of single ORN-types have the ability to influence locomotor preference. Thus, most ORN types are not individually valent. Of the 10 valent ORN types, 6 have previously known roles in valence: two sensors each for pheromones, chemical-threat odorants, and food-aroma molecules. So the past two decades of *Drosophila* olfaction research have already identified a large proportion of strongly valent ORN types. The screen identified four novel valent ORNs (85d, 59 c, 35 a, 47 a) with ligands and ecological roles yet to be determined. Around 60% of pure odorants are valent (*Knaden et al., 2012*), while at least three of the known valent ORN classes (those for geosmin, $CO_2$, and cVA) are specialized to bind specific ligands exclusively. This disconnect between a preponderance of valent odorants (~60% of odorants) and the scarcity of broadly tuned, valent ORN types ($\leq 15\%$) implies that most olfactory valence arises from combined activity in multiple ORN classes (*Mathew et al., 2013*; *Parnas et al., 2013*). This idea is also supported by the Orco activation result showing that broad activation across ~70% of ORNs drives strong attraction.

### Wind and hunger effects on ORN valence are minor

Another goal of this study was to ask whether two contextual factors—wind and hunger—would increase olfactory valence. Prior studies observed that the Orco neurons are either not optogenetically valent or that airflow is essential for valence (*Bell and Wilson, 2016*; *Suh et al., 2007*). However, our experiments showed that olfactory neurons elicit valence without airflow, and that wind has little to no amplifying effect (*Figures 1C and 2*). Note also that even though *Drosophila* larvae behavioral tests are routinely performed in still air, they display both olfactory behavior and optogenetic ORN valence (*Bellmann et al., 2010*; *Hernandez-Nunez et al., 2015*; *Mathew et al., 2013*). Thus, the

overall evidence indicates that *Drosophila* ORNs influence valence behavior even without wind. A second factor, hunger, has also been reported to increase olfactory attraction (*Gelperin, 1971*; *Ko et al., 2015*). However, in the single-ORN screen, starvation had modest effects on valence (*Figure 4*). If anything, valence was somewhat lower for several attractive ORNs (*Figure 4B*). There are at least two plausible explanations as to why starvation did not affect many single-ORN valences. First, the difference in olfactory responses between fed and starved animals might only be pronounced for weaker olfactory stimuli: one study showed that the ethyl acetate (EA) response difference between fed and starved flies declines as the EA concentration increases (*Chakraborty, 2010*). So optogenetic activation might be too strong to observe the starvation effect. Second, ORNs likely have less influence in isolation. Co-activated glomeruli modulate each other via lateral inhibition and excitation (*Groschner and Miesenböck, 2019*; *Huang et al., 2010*; *Shang et al., 2007*; *Wilson, 2008*). As most odors activate multiple ORN types, any hunger effect might require these lateral signals. One study found that five vinegar-responsive ORNs are modulated by hunger only when activated in concert, but not when they are activated in subsets (*Root et al., 2011*). In our results, a notable exception was the hunger switch of Orco valence in females (*Figure 2—figure supplement 2*). The absence of hunger amplification in single-ORN valence—and its presence in female Orco valence—also suggests that the potentiating effects of hunger on olfactory attraction might only operate on multi-ORN stimuli like Orco activation and natural odors. That neither wind nor hunger increased valence of single-ORNs verifies that the majority of single receptor types, on their own, do not convey valence information. Together, the single-ORN results support the idea that most odor-guided locomotion arises from the broad activation of ORNs simultaneously.

## How is ORN information combined?

Many individual odorants bind multiple olfactory receptors, and most natural odors are complex blends that activate receptors broadly, such that odors typically activate multiple ORN classes. A number of groups have constructed statistical models of the relationship between initial layers of olfactory systems and their eventual valent locomotion (*Badel et al., 2016*; *Bell and Wilson, 2016*; *Kreher et al., 2008*; *Kuebler et al., 2012*; *Kundu et al., 2016*; *Meyer and Galizia, 2011*; *Mohamed et al., 2019*; *Riffell et al., 2009*; *Thoma et al., 2014*). The existence of labeled lines indicates that, at least for some odorants and their receptors, ORNs can have a deterministic effect on valence. In a model of *Drosophila* larval olfaction, the weighted summation of ORN activity in just five of 21 receptors could be used to predict odor valence (*Kreher et al., 2008*). In adults, an optogenetic study of ORNs reported that combination valences could be explained by summation and/or max-pooling (*Bell and Wilson, 2016*), further supporting a direct relationship. Summation and max-pooling correspond to special cases of two-way combination weightings of (0.5, 0.5) and (0, 1), respectively; the present analysis found little support for such equal or all-or-none weightings. Rather, the ORN combinations have diverse, intermediate valence weights within the weight space. Moreover, these weights shift as the stimulus intensity increases, and appear to even swap dominance at different intensities. These features suggest complex ORN-channel interactions.

## Possible models of ORN combination

The variability of β values across a range of stimuli indicates that (1) single-ORNs can contribute a dynamic range of weights when co-activated, (2) the experiment captured particular instances of ORN activation from these many possible neural response mechanisms, and (3) that many ORN combinations likely share downstream targets. Therefore, we can cautiously infer that the underlying neural network is multifactorial (*Qiu et al., 2021*) and perhaps context-specific. Several kinds of complex architectures and dynamics could contribute to multi-ORN valence. The increasing dominance of one ORN observed in some of the pairs suggests competitive, antagonistic interactions between channels, such as those mediated by broad lateral synaptic inhibition (*Olsen and Wilson, 2008*) or ephaptic inhibition between neighboring ORNs (*Su et al., 2012*). Alternatively, motor programs associated with valence might only require a subset of channel combinations to be active, with the remainder playing ancillary roles. However, reverse-engineering the exact combination mechanisms through estimated β values would require a larger set of experiments with distinct stimuli. Translating the patterns of β shifts into the specific structure of the neural network carries the risk of over-interpreting the current data set.

## Technical differences between studies

It is relevant to note that several of our conclusions on olfactory valence, notably the nature of ORN-valence combination, diverge from those made in an earlier study (**Bell and Wilson, 2016**). Along with sampling error, it is possible that these discrepancies can be attributed to differences in experimental design and analysis, some of which are summarized in **Supplementary file 2** - Sheet 3. Here, we discuss the different ways the optogenetic valence effect was controlled. As Chrimson was not available at the time, the earlier study used Channelrhodopsin-2 (ChR2), which requires intense blue light (up to 1500 µW/mm$^2$) that adds significant heat and elicits strong responses in the fly visual system (**Supplementary file 2** - Sheet 3). As these stimuli can profoundly alter behavior, especially during a lengthy illumination regime (64 min total), the researchers used genetically blind flies and an infrared laser for compensatory heat. While these technical measures were prudent, they were used largely in place of genetic controls: relative to the experimental animals (N = ~ 2512), the study used few responder UAS controls (N = 88) and no driver Gal4 controls (**Supplementary file 2** - Sheet 3). The earlier study appears to have averaged technical replicates to represent behavioral variation, a procedure that under-reports variation (**Bell, 2016**). The present study dealt with these issues with two changes. First, it was able to make use of the Chrimson channel, which requires lower light intensity and a red wavelength to which the fly eye is less receptive. Second, all non-optogenetic effects were accounted for with balanced experimental, driver, and responder groups (N = ~ 5148 in each group). In our opinion, our approach of testing all three groups in sufficient sample sizes—and using them to calculate effect sizes—enables the exclusion of all confounding influences, including heat, visual effects, and genetic background.

## Limitations of the study

This study incorporates at least four assumptions that potentially limit the generalizability of the findings. First, the use of narrow chambers effectively canalizes fly locomotion into one dimension (**Bell and Wilson, 2016**; **Claridge-Chang et al., 2009**). While this simplifies analysis, it could either alter valence, and/or increase locomotion relative to either broader chambers or behavior in the wild. Second, although the olfactory receptors have been shown to express in specific ORN types (**Couto et al., 2005**; **Fishilevich and Vosshall, 2005**), we cannot rule out that they might express elsewhere in the fly nervous system, so some of the valence results could be the result of activity in several neuron types. Third, optogenetic light is not equivalent to odor stimulation: there may be differences in firing rate, response dynamics, or other features of ORN stimulation that mean these data are not directly generalizable to odor-elicited activity. Fourth, no electrophysiological recordings were made from the optogenetically activated ORNs, so their firing rates relative to odor-elicited activity are unknown; published recordings from optogenetic and odor stimuli suggest an approximately five-fold difference in firing rates in some cases (**Bell and Wilson, 2016**; **Hallem and Carlson, 2006**).

## Do single-ORN-class properties govern valence?

While larval valence has been predicted with a summation model, similar models for olfactory behavior in other systems have not been successful. Studies in various model animals have invoked complex computations to explain olfactory valence (**Duchamp-Viret et al., 2003**; **Kuebler et al., 2012**; **Kundu et al., 2016**; **Meyer and Galizia, 2011**; **Riffell et al., 2009**; **Shen et al., 2013**; **Silbering and Galizia, 2007**). In adult *Drosophila*, an analysis of the physiological and behavioral responses to 110 odors found that ORN activities and odor valence have no linear correlation (**Knaden et al., 2012**), suggesting that valence determination could arise from a downstream computation, for example in the antennal lobe, where incoming ORN activity patterns and outgoing projection-neuron activity patterns are dissimilar (**Groschner and Miesenböck, 2019**). Indeed, the physiological-activity patterns in projection neurons have been reported to be at least partially predictive of odor valence (**Badel et al., 2016**; **Knaden et al., 2012**; **Parnas et al., 2013**), leading to the idea that the antennal lobe extracts valence features. To examine this, we modeled published ORN activity and valence data sets, finding that linear models of ORN activity data could account for a minor fraction of olfactory-behavior variance, and this was not substantially improved with single-class ORN valence information. All the observations in the present study point to a minimal role for simple pooling rules, and support the idea that odor valence is primarily governed by complex circuit dynamics.

## Materials and methods

### *Drosophila* strains

Flies were raised on fly medium (*Temasek Life Sciences Laboratories, 2018*) at 25 °C in a 12 hr light:12 hr dark cycle. For optogenetic experiments, the flies were kept in the dark and reared on fly food supplemented with 0.5 mM all-trans-retinal (Sigma-Aldrich, USA) for 2 days prior to experimentation. For starvation experiments, the flies were reared on 2% agarose for 12–18 hr prior to the assay. Wild type flies were cantonized $w^{1118}$; all the *ORx-Gal4* and *UAS-CsChrimson* strains were obtained from the Bloomington *Drosophila* Stock Center (USA) (*Supplementary file 2* - Sheet 4) (*Couto et al., 2005*; *Dobritsa et al., 2003*; *Klapoetke et al., 2014*; *Vosshall et al., 2000*).

### Optogenetic preference assay

The wind- and light-induced self-administration response (WALISAR) assay was conducted as follows. Two rectangular assemblies (11.5 × 14.5 × 0.3 cm) were cut from acrylic sheets; each assembly contained 26 chambers (50 × 4 × 3 mm), herein referred to as WALISAR chambers. Airflow inlets and outlets were milled into the ends of each chamber. Optogenetic illumination was achieved using LEDs [LUXEON Rebel LEDs on a SinkPAD-II 10 mm Square Base; red (617 nm), green (530 nm), blue (470 nm), each equipped with lenses (17.7° 10 mm Circular Beam Optic)] and attached to heatsinks located above the arena on both sides at a ~ 45° angle. The LEDs were grouped by color and were powered by 700 mA BuckPuck drivers. Custom instrumentation software (CRITTA) was used to control the intensity and timing of the LEDs throughout the experiments. To achieve a half-dark/half-lit arena for the optogenetic choice experiments, two black acrylic shields were placed between the arena and the LEDs on each side and were adjusted to cast shade on either half of the chamber. By switching the LEDs on either side, the half of the arena that was lit could be alternated. The temperature difference between the dark and lit halves of the WALISAR chambers that could arise from the LED illumination was measured for the duration of a whole experiment using thermocouples, and found to be a negligible ~0.3 °C (https://doi.org/10.5281/zenodo.4545940). Compressed air was connected to the airflow inlets of the arena, with an intervening stopcock valve (Cole-Parmer) to modulate the flow rate. The air flow in all experiments using wind was 35 cm/s, as described in a previous study (*Bell and Wilson, 2016*). The airflow was measured before every experiment with an airflow meter (Cole-Parmer).

Flies were collected 2–3 days before experiments and cold anesthesia was administered immediately prior to their transfer into the WALISAR chambers. Unless otherwise stated, each experiment typically used sample sizes of N = ~ 104 Gal4 and UAS controls and N = ~ 52 test flies. Transfer to individual chambers took around 5 min. A single experimental cycle consisted of: acclimatization of the flies for 30 s; illumination of the left half of the arena for 45 s; no illumination for 30 s; illumination of the right half of the arena for 45 s; and no illumination for 30 s. The chambers were manually tapped against the incubator base before each cycle. Each group of flies was tested in six conditions comprising three light intensities (14, 42, 70 µW/mm2), each with and without wind (35 cm/s) (on/off), totaling 12 steps (S1–S12) (*Figure 1—figure supplement 1A*). The total experiment duration was 180 s × 12 = 2160s = 36 min. The light intensities were measured with a thermal power sensor (Thorlabs S310C) connected to a power and energy-meter console (Thorlabs PM100D). The flies were recorded with an AVT Guppy PRO F046B camera fitted with an IR bandpass filter, which was positioned on the top of the arena. The camera was connected to a computer running custom LabVIEW software (CRITTA), which was used to determine the flies' head positions.

### WALISAR protocol and validation

Given that the ordering of the experiments in repeated-measure designs can affect outcomes, the WALISAR protocol was tested with two sets of experiments performed on *Orco-Gal4> UAS-CsChrimson* flies, in an ascending or descending light-intensity order (*Collie et al., 2003*; *Howitt and Cramer, 2007*; *McCall and Appelbaum, 1973*). The responses in the two orders were similar for eight of the 12 epochs, being different only for S2, S3, S10, and S11. The difference was due to the weak valence responses in the second- and third-order epochs: S2 and S3 produced lower effects in the ascending order (*Figure 2—figure supplement 1E*), S10 and S11 produced lower effects in the descending order (*Figure 2—figure supplement 1F*). The underestimation of the second- and third-order epochs had little effect on the overall interpretation of the results because eight of the

epochs generated similar results. Moreover, even a hypothetical extreme case in which the valence is overlooked in the second- and third-order epochs would result in a false negative, rather than a false positive. As such, the experimental order was concluded to not have a major effect on the WALISAR results. Additionally, the effect sizes in the first epoch tended to be smaller than the second epoch when the light was downwind of the air flow (*Figure 2—figure supplement 1*); to eliminate this bias, only second epochs were used for further data analysis.

One concern is that the activity rates and temporal structure of optogenetic stimulation are likely different from direct odor stimulation. To mitigate this, we conducted a study of the effect of optogenetic temporal structure, finding that—while this is a relevant concern—continuous illumination is a more conservative method (*Tumkaya et al., 2019*). We also benchmarked behavioral responses for the Orco neurons against results from a prior study that performed physiological recordings and used a different temporal structure (*Bell and Wilson, 2016*), finding that the WALISAR protocol has comparable sensitivity (*Figure 1—figure supplement 3*).

## WALISAR data analysis: the wTSALE metric

Fly-position data were analyzed with custom Python scripts. The valence of each fly was measured in terms of how much time it spent in the light after first encountering one of the lit zones. Specifically, the first frame in which a fly entered the lit zone was considered the start of the test session; after this initial light encounter, the amount of time spent in the dark was subtracted from time spent in the light and finally divided by the total amount of time. This metric is designated 'Time Spent After Light Encounter' (TSALE). The duration between the first light discovery and the end of the light epoch varied across individual files, from never discovering the light (0 s) to being exposed to the light at the start of the epoch (60 s). As such, each fly's TSALE score was weighted by the post-light duration, termed 'weighted Time Spent After Light Encounter' (wTSALE). This weighting was achieved by multiplying the TSALE with the ratio between the remaining time and the full duration of the test epoch. The wTSALE score was calculated for each fly and then averaged for the control and test genotypes. To calculate the effect sizes, responder and driver controls were pooled into a single control group. The mean difference (Δ) between the pooled-control and test groups was taken as the effect size (ΔwTSALE). The Δ distributions and 95% bootstrapped confidence intervals (CIs) were calculated using the DABEST Python package (*Ho et al., 2019*), and presented in the results in the following format: "Δ [95 CI lower bound, upper bound]."

## Overview of the Empirical Bayes method

Empirical Bayes (EB) is a method for statistical inference using Bayesian hierarchical models. While relatively unknown in behavioral genetics, EB is widely used to filter omics data, having been originally developed for microarray data, and currently routinely used for, among other applications, mass spectrometry proteomics (*Koh et al., 2019*). This approach is intrinsically connected to the traditional analysis workflow based on hypothesis testing with multiple-testing correction, but with a major difference (*Efron and Tibshirani, 2002*). The hypothesis-testing approach is solely based on tail probabilities under the null hypothesis, and does not consider true signals. The key advantage of EB is that it explicitly models distributions of both the noises (null hypotheses) and the true signals (alternative hypotheses). Thus, for the analysis of a genetic screen, the key difference of this approach (compared to the conventional hypotheses testing-based analysis) is that we evaluate the significance of response levels across all ORNs simultaneously, modeling their distributions as a mixture of a random noise component and a real signal component. Empirically speaking, when the true effect sizes are modest, this typically increases the sensitivity of detecting mild effect sizes. Further, the mixture modeling produces a confidence score, that is the posterior probability of true signal from the underlying Bayesian model, which is the complement of the local false discovery rate (*Efron, 2010*).

## Data analysis with Empirical Bayes

To draw conclusions from multiple experiments (performed at one of three light intensities) for a given ORN, the dimension of the data was reduced into one summary statistic ($D$) for each ORN, as follows *Efron et al., 2001*:

$$D_i = M_{exp} - \left( \left( M_{ctrl1} + M_{ctrl2} \right) * 0.5 \right),$$

where $M$ is the mean of the wTSALE score for experimental and control flies. Then, $D$ was used to calculate a $Z$ score:

$$Z_i = D_i / (\alpha_0 + S_i),$$

where $D_i$ is the effect size of each ORN, $S_i$ is the standard deviation, and $\alpha_0$ is 90th percentile of all the $S$ values (*Tusher et al., 2001*).

After calculating a vector of $Z$ scores, the EBprot software package (*Koh et al., 2015*) was used to apply the empirical Bayes method to differentiate true behavioral changes from noisy observations through direct estimation of the respective distributions. The empirical Bayes method directly models the valence Z scores as random variables from a two-component mixture, where the noise component is modeled as Gaussian distribution and the signal component (true behavioral changes) is modeled by a non-parametric distribution (*Efron, 2010*). In effect, the method borrows statistical information across the ORNs to learn the data generating distributions of the null and alternative hypotheses. As the inferential method is based on the Bayesian framework, the model fit naturally yields the posterior probability of true change of each ORN as well as the false discovery rate (FDR) associated with any posterior probability threshold. The true signal component is subsequently divided into positive and negative sub-components depending on the sign of the Z-scores. From this three-component mixture, a signed probability is derived to indicate the likelihood that the valence score came from each of the three candidate distributions (negative, neutral, and positive valence). Prior to calculating the signed probabilities, EBprot removes outliers from the valence distributions (*Koh et al., 2015*). Effect sizes associated with a < 25% FDR were considered for further analysis.

## Modeling the valence responses of ORN combinations

The ORN-combo valences were modeled using three functions, each widely used in neural network research for input pooling: summation, max-pooling, and min-pooling. Summation simply sums two individual valence values, while max-pooling and min-pooling return the larger or smaller absolute value of the two components, respectively (*Goodfellow et al., 2016*), as follows:

$$Summation = (i_0 + i_1)$$

$$Max - pooling = max (i_0, i_1, key = absolute value)$$

$$Min - pooling = min (i_0, i_1, key = absolute value)$$

where $i_0$ and $i_1$ are real numbers.

For example, given $i_0 = -2$ and $i_1 = +1$, the summation, max-pooling, and min-pooling functions would return -1, -2, and +1, respectively.

## Correlation and agreement analyses

Linear regressions were performed using the SciPy library in Python (*Jones et al., 2001*). Bootstrapped CIs for the coefficient of determination ($R^2$) were calculated using the scikits-bootstrap package in Python (*Evans, 2019*). Bland-Altman plots were generated using custom scripts using the matplotlib and seaborn libraries in Python (*Bland and Altman, 1999*; *Hunter, 2007*; *Waskom et al., 2017*).

## ORN interaction analyses

Custom R scripts were used to perform statistical inference on multiple linear regression models characterizing the association between single-ORN classes and ORN-combo. Single-ORNs (ORN1 and ORN2) were used as predictors, while the ORN-combo phenotype was the dependent variable, as follows:

$$ORN - combo = \beta_0 + \beta_1 \times ORN1 + \beta_2 \times ORN2$$

From the original data, 10,000 bootstrap samples of the single-fly data were drawn from each group (ORN1, ORN2, and ORN-combo) and the flies were ordered by their valence levels and paired into trios of ORN1, ORN2, and ORN-combo. In each bootstrap sample, a multiple linear regression model was fitted, which revealed the probabilistic distributions of the beta weights ($\beta_1$ and $\beta_2$).

## Predictive modeling

We considered that the available published data has a disproportionate ORN to odor ratio (23–110), with the possibility of internal correlation. Prior to building a valence prediction model, we performed exploratory analyses on the datasets: Pearson correlation among the ORNs, hierarchical clustering, and partial least squares discriminant analysis (PLS-DA). All three analyses were conducted using the pandas and scikit-learn libraries in Python (*McKinney, 2010*; *Pedregosa et al., 2011*). The eight latent variables (LVs) calculated by the PLS-DA analysis were used to train the linear and non-linear models using the scikit-learn library in Python (*Pedregosa et al., 2011*). The performance of the models were evaluated by using the adjusted-$R^2$ and the root-mean-squared error (RMSE). The RMSE values were calculated using a shuffled-split ten-fold cross-validation scheme in Python's scikit-learn library (*Pedregosa et al., 2011*). To weight the prediction models, we calculated the Pearson correlation coefficients between Hallem and Carlson's firing rates and our single-ORN wTSALE valence scores of the 23 ORNs for all 110 samples. So that the odors with higher correlation play a more important role in the model fitting, absolute value of these correlations were then used to re-weight the odor samples in the regression model.

## Re-analyzing previous valence studies

The relevant studies were downloaded in pdf format, and the data of interest were extracted by using the measuring tool in Adobe Acrobat Pro (Adobe Systems USA). The extracted values for control and experimental groups were then used to calculate the standardized effect size, Cohen's *d* (*Cumming and Calin-Jageman, 2016*). If technical replicates were used to calculate statistics, the effect sizes were corrected accordingly (*Supplementary file 2* - Sheet 3).

## Code Availability

All of the data generated by this study are available to download from Zenodo (https://doi.org/10.5281/zenodo.3994033). The code is available at (https://github.com/ttumkaya/WALiSuite_V2.0; *Tumkaya, 2022*; copy archived at swh:1:rev:35ca421e06223bc2d5f167783c3029b2a8240a85).

## Acknowledgements

We thank Joses Ho, other members of the lab and TT's thesis committee for helpful feedback. We thank Jessica Tamanini of Insight Editing London for a critical reading of the manuscript before submission. We thank Franz Anthony for the assay illustration. Funding: The authors were supported by grants from the Ministry of Education (grant numbers MOE2013-T2-2-054 and MOE2017-T2-1-089) awarded to ACC. TT was supported by a Singapore International Graduate Award from the A*STAR Graduate Academy. HC was supported by grant MOE-2016-T2-1-001 from the Singapore Ministry of Education and NMRC-CG-2017-M009 from the National Medical Research Council. The authors received additional support from Duke–NUS Medical School, a Biomedical Research Council block grant to the Institute of Molecular and Cell Biology, and grants from the A*STAR Joint Council Office (grant numbers 1,231AFG030 and 1,431AFG120) awarded to ACC.

## Additional information

### Funding

| Funder | Grant reference number | Author |
| --- | --- | --- |
| Agency for Science, Technology and Research | AGA-SINGA | Tayfun Tumkaya |
| Agency for Science, Technology and Research | Block grant | Tayfun Tumkaya James Stewart Hyungwon Choi Adam Claridge-Chang |

| Funder | Grant reference number | Author |
|---|---|---|
| Ministry of Education - Singapore | MOE2013-T2-2-054 | Tayfun Tumkaya<br>James Stewart<br>Adam Claridge-Chang |
| Ministry of Education - Singapore | MOE2017-T2-1-089 | Tayfun Tumkaya<br>James Stewart<br>Adam Claridge-Chang |
| Ministry of Education - Singapore | MOE-2016-T2-1-001 | Hyungwon Choi |
| National Medical Research Council | NMRC-CG-2017-M009 | Hyungwon Choi |
| Duke-NUS Medical School | Block grant | Adam Claridge-Chang |
| Agency for Science, Technology and Research | JCO-1231AFG030 | James Stewart<br>Adam Claridge-Chang |
| Agency for Science, Technology and Research | JCO-1431AFG120 | James Stewart<br>Adam Claridge-Chang |
| Ministry of Health | MOE2019-T2-1-133 | Adam Claridge-Chang<br>James Stewart |

The funders had no role in study design, data collection and interpretation, or the decision to submit the work for publication.

## Author contributions
Tayfun Tumkaya, Conceptualization, Data curation, Investigation, Methodology, Resources, Software, Visualization, Writing - original draft, Writing - review and editing; Safwan Burhanudin, Asghar Khalil-nezhad, Investigation; James Stewart, Formal analysis, Methodology, Software, Software: LabView; Methodology: Instrumentation; Hyungwon Choi, Formal analysis, Methodology, Software, Software: R, Supervision, Writing - review and editing; Adam Claridge-Chang, Conceptualization, Funding acquisition, Methodology, Project administration, Supervision, Writing - review and editing

## Author ORCIDs
Tayfun Tumkaya ⬡ http://orcid.org/0000-0001-8425-3360
Hyungwon Choi ⬡ http://orcid.org/0000-0002-6687-3088
Adam Claridge-Chang ⬡ http://orcid.org/0000-0002-4583-3650

## Decision letter and Author response
Decision letter https://doi.org/10.7554/eLife.71238.sa1
Author response https://doi.org/10.7554/eLife.71238.sa2

# Additional files

## Supplementary files
• Supplementary file 1. ORN types with known innate valence. The innate responses of 24 types of ORNs that have been reported at time of writing (January, 2020). The valence response, experimental assay, stimulus type, sex, and the developmental stage of the flies varied across studies. The valence column shows the response direction produced by the respective ORN: negative (-), positive (+), or indifferent (o). The assay column presents the nature of the assay used in the experiments: oviposition (place preference for egg-laying in female flies), two-choice (any assay by which the flies are presented with a choice to activate the ORN), locomotor (the assay in which motor behavior of the larvae is used to deduce valence). The stimulus column shows the type of the stimulus applied on the ORN: olfactogenetics (geosmin on ORNs that ectopically express the Or56a receptor), optogenetic (light stimulus on genetically modified ORNs), odor. In the sex column, F, M, M/F, and N/A indicate female, male, both male and female, and information that is not available, respectively. The stage column presents the developmental stage of the animals used in the experiment: larva, adult, and both (larva and adult).

• Supplementary file 2. Sheet 1: Ten-fold cross-validation of ORN–preference prediction models. RMSE scores were calculated with cross-validation for four models of ORN activity and odor

preference. All RMSE scores are close to the standard deviation of the dependent variable (σ = 0.23). Sheet 2. Ten-fold cross-validation of single-ORN-valence-weighted prediction models. Sheet 3. A comparison of the experimental designs of two optogenetic studies investigating ORN-valence behaviour. Sheet 4. A list of ORN-Gal4 lines used in the study.

• Transparent reporting form

## Data availability

Data and code availability: All of the data generated by this study are available to download from Zenodo (https://doi.org/10.5281/zenodo.3994033). The code is available at https://github.com/ttumkaya/WALiSuite_V2.0 copy archived at swh:1:rev:35ca421e06223bc2d5f167783c3029b2a8240a85.

The following dataset was generated:

| Author(s) | Year | Dataset title | Dataset URL | Database and Identifier |
|---|---|---|---|---|
| Tumkaya T, Burhanudin S | 2020 | Dataset for: Majority of olfactory-receptor neurons have individually neutral effects on behavior | https://zenodo.org/record/3994033 | Zenodo, 10.5281/zenodo.3994033 |

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
