## [Editor Report]

Olfactory coding is still an open question in neuroscience. Therefore, this paper is of potential interest to a broad audience of neuroscientists. It undertakes a thorough investigation of how olfactory sensory neurons drive avoidance or attraction in flies and also addresses how combinations of active ORNs can become behaviorally meaningful. It has great potential value for clarifying how animals map sensory input to valence.

---

## [Decision Letter]

**Decision letter after peer review:**

Thank you for submitting your article "Most primary olfactory neurons have individually neutral effects on behavior" for consideration by *eLife*. Your article has been reviewed by 3 peer reviewers, and the evaluation has been overseen by a Reviewing Editor and Piali Sengupta as the Senior Editor. The following individual involved in review of your submission has agreed to reveal their identity: Matthew C Smear (Reviewer #2).

We appreciated the quality, extent, and importance of this work. While doing so, we had a few concerns that we think the authors should be able to address. They are listed below.

1. Anaesthesia: We were concerned about the short recovery period post cold anesthesia and prior to the behavioural assay. Since cold anesthesia is known to have effects on behaviour, could the authors please demonstrate that a longer duration of recovery doesn't alter their findings of neutral ORN valence?

2. wTSALE: We were concerned that this method of weighting used by the authors may be obscuring real behavioural phenomenon and therefore masking valence. Could the authors please revisit this? Providing more traces of the different response types – attraction, avoidance, weak responses, etc. – would also be helpful.

3. ORN combinations: One of the key points of this manuscript is what it tells us about the rules by which ORN combinations work. While the authors show what their study rules out, we felt that they fall short of discussing what might be occurring instead. So, could the authors please include some discussion around this point?

In this section we also recommend incorporating SupFig8 into the main figure 5. It possible that different ORN pair use different interaction rules and the grouped analysis in figure 5 would mask this. Sup Fig8 is more informative in this regard.

4. Statistics: The statistics in this manuscript are quite involved. We recognise that the authors are promoting the use of Empirical-Bayes methods for statistical inferences. Since this is not commonplace, could the authors please incorporate an intuitive explanation about Empirical Bayes, its assumptions, and why it's better suited to the analysis? We think this will greatly improve accessibility of this manuscript, and therefore its impact.

5. Comparison with the Bell and Wilson study: Could the authors please include the number of ORN pairs tested in the B and W study and their own (28 and 7)? With respect to the stimulation conditions listed in this table (Supp Table 4), we assume that the authors' count of 6 conditions is because they are including with and without airflow for their 3 light intensities. In this case, the B and W study should be listed as 16. Alternatively (indeed preferably), the two studies should be listed as 8 and 3 respectively.

*Reviewer #1 (Recommendations for the authors):*

Figure 1A I cannot tell the direction of airflow in the corridors. The air port is only shown on one side, does that mean airflow is unidirectional? Where is the exhaust?

I would prefer a more schematic/conceptual drawing of the arena than this quasi-realistic one where the main feature that pops out to me are the flies themselves. I would prefer that the drawing conveys the technical details needed to evaluate what the flies experience in the assay.

Fig1F The y-axis label wTSALE should be swapped for a term with some intrinsic meaning. There isn't even any basic description of what wTSALE means in the Results section, the reader has to go to the Methods. I think it would be helpful for the reader to understand the assay more clearly if the full description is in the Results. It is basically the proportion of time the flies spend outside the light, with the clock starting only after the fly has experienced the light the first time. That's pretty easy to understand and the y-axis label could be % time outside light' or even 'preference light' for positive values and 'preference dark' for negative values without criminally oversimplifying the measurement (IMO).

Also I am not convinced the weighting the authors use (i.e. wTSALE vs TSALE) is really justified. Essentially they are trying to control for shorter sampling periods of the fly's behavior in an extremely simple linear way. That implies that short sampling periods may not be representative – is it fair to simply weight those down so the score goes closer to zero since that actually indicates a lack of preference? Basically since any value on their y-axis carries meaning, it seems unfair to weight some points down simply because sampling wasn't extensive enough. Why not just require some minimal time window for flies to have experienced the light (i.e. know what they are choosing) and look at the overall proportion of time in light vs dark?

P.10 "This result indicates that wind has essentially no impact on ORN-elicited behavior in walking flies" this should read 'single-ORN-elicited behavior' since this is all the authors tested. ORN-elicited behavior could be read as ORN activation in general i.e. odor-based activation, where there is likely an effect of wind at least in some assays.

Figure 5: The authors analyze all ORN pairs together to test whether they summate/max pool/min pool but prior work (Bell and Wilson) showed that some pairs summate while others max pool, which would confound the style of grouped analysis in Figure 5G-I.

Additionally since only 7 combinations were used and only 3 intensity levels, this figure is the weakest part of the paper, which up to this point has been extremely extensive. It also makes the first entry in Table 4 (Number ORNs tested = 45 vs n=8 for Bell and Wilson) unfair since Bell and Wilson actually looked at all combinations among 8 ORNs.

Can the authors discuss more about how the dominant β value can flip as the stimulation intensity increases? How would that work in terms of neural activity in the biological network? Also, what does the diversity of β values imply about biological network, does it potentially correspond to different weights on different downstream targets?

Ending on a negative result is a little disappointing – one positive point the authors could make is that (with one exception) the ORN combinations all transition towarrds more max pooling at higher stimulation rates. This suggests an competitive interaction between channels, which is easy to imagine. Undoubtedly it is complex with different downstream targets having different rules, but this is one fairly consistent trend.

I should say that I found Bell and Wilson more convincing because they examine interactions for each ORN pair over a wider range of spike rates. Here there are just three points for comparison, and when my eyes look at Figure 5C-E it seems that there is not a lot of difference between the three interaction modes.

Finally, the authors should somehow incorporate FigS8 into the main text since I'm sure that the interaction mode depends on the pair of ORNs being examined.

*Reviewer #2 (Recommendations for the authors):*

– I need to understand the raw data better. What are the flies actually doing here? In 1B, the example fly seems to be walking back and forth at about 0.1 Hz. Is this representative of the population? Do the flies ever not move at all? How is this outcome dealt with? The methods mention that Empirical Bayes has a principled way of excluding outliers. What is that way? When the fly's path enters into the illuminated region, it seems to immediately stop and walk back to the opposite wall, and then on its next two cycles it stops before entering the illuminated region. Is this because the light spreads or does the fly remember where it hit the light before? What happens when the illumination occurs when the fly is already on the illuminated side? The effect of Gr66a>Chr (1F) is much larger than any of the OR effects. What does a weaker avoidance response look like? What does an attraction response look like? The mean +/- 95% CI plots of 1C-E do not answer these questions. More individual animal trajectories and population occupancy heat maps would help a lot. Exclusively compressing the data to the one wTSALE number may well be obscuring worthwhile features of the behavior. With a richer characterization of the behavior, it might be possible to reduce the sample size and simplify the statistics.

– The statistical methods are unusual and seem unnecessarily complicated (at least to me). Further, why these were used instead of something more conventional? Readers (at least this reader) would benefit greatly from clear language giving an intuition for how Empirical Bayes works, what are its assumptions, and why it is superior to more conventional, easier-to-understand methods.

– The distribution of wTSALE in Figure 2 F and G is striking. In these plots, including the controls, there is a large mode at wTSALE=1. This mode is not apparent in the distributions of 1F. Why are control flies so much more attracted to light in these experiments? How does Empirical Bayes deal with non-Gaussian distributions?

– What direction does the wind flow through the chamber? It appears to run perpendicular to the illumination axis. Could this matter? Does wind itself impact the locomotion of the flies? Since only δ-wTSALE is shown, it seems possible that wind may affect the behavior in a way that would obscure an effect. Here again it would be helpful to show more of what the flies are actually doing.

– The authors invoke "complex circuit dynamics" to explain the results of the combined-receptor experiments. I'm not sure what the authors mean here. "Dynamics" implies that time-dependent processes determine valence. If this were the case, these experiments would show no effect, since the stimuli don't recapitulate the dynamics of odor-evoked ORN activity. The discussion in a recent paper by Ron Yu's group (Qiu et al., 2021; Current Biology), deals with the non-labelled-line-ness of the mouse olfactory system in a thoughtful way. A similar discussion would benefit this paper as well.

*Reviewer #3 (Recommendations for the authors):*

1. The authors use a large array of GAL4 driver lines that they claim cover only the relevant ORN type. However, for most of these lines this was not examined. Although in the past such lines were used for behavior experiments, recent studies are much stricter with the use of driver lines. Many studies have demonstrated that even expression in a single neuron (other than the target neurons) either in the central brain or in the VNC can affect behavioral results. The authors therefore must show that the lines used in this study only label the target neurons either by providing adequate citations or by examining this directly with confocal stacks of both whole brain and VNC.

2. The authors do not show the relevance of their optogenetic activation of ORNs to odor activation of ORNs. Previous studies have shown that optogenetic activation of ORNs generates a firing rate of approximately 30 Hz (Bell and Wilson, 2016; Fox and Nagel, 2021). In contrast, ORNs can reach firing frequencies of up to 250 Hz in response to odors (Hallem and Carlson, 2006). In addition, ORNs show temporal dynamics, whereas I presume that the continuous illumination generates a more uniform response. The authors briefly discuss this in the methods section. They claim that "continuous illumination is a more conservative method (Tumkaya et al., 2019)". However, the same authors claim in their Tumkaya et al., 2019 manuscript that "These results suggest that neither stimulation type is necessarily superior to the other: static- or pulsed-light stimulation can capture more of the native responses than the other in inducing olfactory behavior, depending on the neuronal type". The authors also claim that "We also benchmarked behavioral responses for the Orco neurons against results from a prior study that performed physiological recordings and used a different temporal structure (Bell and Wilson, 2016), finding that the WALISAR protocol has comparable sensitivity (Figure S3)". The fact that both optogenetic activations has similar behavioral results does not imply any relevance to an olfactory cue.

My main concern is that the current optogentic stimulation probably activates ORNs relatively weakly, thus mimicking low odor concentration. As low odor concentrations elicit in many cases only weak behavioral responses it is more than possible that the lack of behavioral effect is just due to "low concentration" and not an indication to the actual role of each ORN.

Taken together, I think the authors should go the extra mileage and show some relevance to olfactory stimuli.

3. The authors own data raises potential problems with their approach. Some of the ORNs that are classified as driving aversion or attraction seem to change valence value they induce with the light intensity. For example, the authors report Or42b to drive attraction in agreement with published literature. However, at the strongest light intensity it is actually neutral. Similarly, the authors report Or85d to drive aversion. However, at the strongest light intensity it is also neutral. So, are these ORs "neutral"?

4. The authors test a number of previously suggested linear models and find that they do not predict how two-ORN odor valence emerges from single-ORN valence. However, linear models were shown to be insufficient to predict odor valence (Badel et al., 2016). It is thus not surprising that these linear models failed.

5. The authors use two databases, one of odor responses (Hallem and Carlson 2006) and one of behavioral responses (Knaden et al., 2012) along with a linear model to try and predict odor valence from ORN activity. However, as mentioned above linear models are not adequate for describing the relation between ORN activity and Odor valence. Furthermore, I think the Knaden et al., database is a wrong database to use in this context. Knaden et al., used a trap assay. In this assay, flies are captured in the trap after a single entrance to the odor source. Thus, exploratory behavior, in which flies examine the odor and then can decide to avoid it, cannot occur, and this assay is expected to be biased towards reporting odors as attractive. Indeed, this was the case in the Knaden et al., database in contrast to other published results. This database was suitable for the claim raised by Knaden et al., that looked only at the most aversive and attractive odors, but it cannot be used to try to predict any odor valence.

6. The authors used cold anesthesia just prior to loading the flies to the chambers and only 30 second acclimation following the cold anesthesia. However, cold anesthesia is known to have effects on behavior, increasing response time, reducing locomotion and reducing overall responses (just a few examples, Barron, 2000; MacMillan et al., 2017, Trannoy et al., 2015). I think most studies today try to avoid cold anesthesia just before the experiment. My concern here is that the lack of effect for most ORNs, may arise from general behavior impairment. Can the authors give a few examples from the neutral ORNs without cold anesthesia?

7. The authors conclude that: 1. "the majority of primary olfactory sensory neurons have neutral behavioral effects individually". This conclusion (as mentioned above) is definitely correct for the optogenetic activation, but its relevance to odor valence is questionable. Furthermore, Badel et al., 2016 already demonstrated with actual odor stimuli that “We find that the behavior is accurately predicted by a model summing normalized glomerular responses, in which each glomerulus contributes a specific, small amount to odor preference.” Thus, the novelty of the current study is not large.

Their second conclusion is that “olfactory sensory neurons…participate in broad, odor-elicited ensembles with potent behavioral effects arising from complex interactions”. I agree with them that olfactory coding is complex. However, they did not show any actual odor responses to support their claim, neither did they provided even one complex mechanism. I think that stating that olfaction is complex is just not enough.

8. To my understanding the order of the β coefficients can affect the interpretation of the data. However, I could not find a reference for this in the methods. Can the authors please elaborate on this?

---

## [Author Response]

We appreciated the quality, extent, and importance of this work. While doing so, we had a few concerns that we think the authors should be able to address. They are listed below.1. Anaesthesia: We were concerned about the short recovery period post cold anesthesia and prior to the behavioural assay. Since cold anesthesia is known to have effects on behaviour, could the authors please demonstrate that a longer duration of recovery doesn’t alter their findings of neutral ORN valence?

We appreciate this concern, however there are three lines of evidence that suggest the brief ice anesthesia is not a major cause of effect-size underestimation.

First, fly loading takes around 5 min and the full experimental duration is 36 min, meaning that, for later epochs, the effective total-recovery durations are over 30 min. Thus, even if earlier epochs were underestimating valence, the latter epochs are less likely to be affected.

Second, when compared with the literature, the positive controls are concordant. To benchmark our method, we used a quantitative literature review with standardized effect sizes of two reference lines: Gr66a-Gal4 and Orco-Gal4 (Figure 1-S3). It shows that our assay yields valence that is very similar to previous studies.

Comparing maximal scores, the comparisons of these lines include a 20% decrease (Gr66a), a 10% increase (Orco-Gal4 with wind), and a 5-fold increase (Orco-Gal4 in no wind). This shows that the overall protocol with the brief cold anesthesia does not systematically underestimate valence when compared with published protocols.

Third, the unbiased screen turned up a majority of hits that were already reported in the literature as being valent ORN types. Discussed at greater length below, this also indicates that there is nothing especially defective about the ice protocol.

Action taken: We added text to Methods to clarify the total duration. “The total experiment duration was 180 s × 12 = 2160s = 36 min.”

We added a row to the comparison in Supplementary File 2 – Sheet 3 to include that Bell and Wilson used aspiration, while we used cold anesthesia.

2. wTSALE: We were concerned that this method of weighting used by the authors may be obscuring real behavioural phenomenon and therefore masking valence. Could the authors please revisit this? Providing more traces of the different response types – attraction, avoidance, weak responses, etc. – would also be helpful.

We appreciate this concern and have addressed it by showing a case study of *Or67d-Gal4* (Author response image 1) and several other drivers (Author response images 2–7) finding that the choice of metric has a modest impact on the overall valence outcome. For Or67d, the majority of the flies discover the optogenetic light within the first ~5 seconds of a WALISAR experiment (Author response image 1). The full-epoch metric conserves more preference information than a short-epoch cutoff, but there still are a few flies whose first light encounter does not occur until much later in the epoch (Author response image 1). For example, a fly that encounters the light in the final three seconds of the test will give less reliable preference information. To account for this difference, we linearly weighed the preference scores by the percentage of epoch that the flies actually experienced. So, wTSALE does not underestimate valence, but rather is an unbiased and robust metric that is fairly similar to other metrics.

**Author response image 1. sa2fig1:** wTSALE presents Or67d-induced behaviour more accurately than commonly used metrics in the fieldA. An estimation plot presents the individual preference (upper axes) and valence (lower) of flies with activatable Or67d+ neurons in the WALISAR assay. In the upper panel, each dot indicates a preference index (PI) for an individual fly that was calculated by using the last 3 second of the epoch: *w1118; Or67d-Gal4* and *w1118; UAS-CsChrimson* flies are colored blue; and *Or67d-Gal4 > UAS-CsChrimson* flies are in orange. The mean and 95% Cis associated with each group are shown by the adjacent broken line. In the bottom panel, the black dots indicate the mean difference (ΔwTSALE) between the relevant two groups: the valence effect size. The black whiskers span the 95% Cis, and the orange curve represents the distribution of the mean difference. B. An estimation plot presents the individual preference (upper axes) and valence (lower) of flies with activatable Or67d+ neurons in the WALISAR assay that were calculated by using the last 5 seconds of the experiment. C. Preference index (PI) for each fly that used only the final 10 seconds of the experiment. The color-code, layout, and statistics are the same as Panels A and B. D. Preference index (PI) for each fly when the whole epoch (45 seconds) was included into the analysis. The color-code, layout, and statistics are the same as Panels A and B. E. The individual preference (upper axes) and valence (lower) of flies by the wTSALE metric. F. A scatter plot shows how long it takes for each fly to encounter the optogenetic light once it is switched on. The median and 95% Cis associated with each group are shown by the adjacent broken line.

**Author response image 2. sa2fig2:** Or67d neuron activation is attractive to flies A-C-E. Trace plots representing the mean location of controls (*w^1118^; Or67d-Gal4* and *w^1118^; UAS-CsChrimson*), and test flies (*Or67d-Gal4 > UAS-CsChrimson*) throughout an experiment at 14, 42, and 70 μW/mm2 light intensities. The blue and orange ribbons indicate 95% CIs for the control and test flies, respectively. Second epochs were used to calculate the preference scores (shown in black rectangle). B. An estimation plot presents the individual preference (upper axes) and valence (lower) of flies with activatable Or67d+ neurons in the WALISAR assay. In the upper panel, each dot indicates a preference score (wTSALE) for an individual fly: *w^1118^; Or67d-Gal4* and *w^1118^; UAS-CsChrimson* flies are colored blue; and *Or67d-Gal4 > UAS-CsChrimson* flies are in orange. The mean and 95% CIs associated with each group are shown by the adjacent broken line. In the bottom panel, the black dots indicate the mean difference (ΔwTSALE) between the relevant two groups: the valence effect size. The black whiskers span the 95% CIs, and the orange curve represents the distribution of the mean difference. D. A scatter plot shows how long it takes for each fly to encounter the optogenetic light once it is switched on. The median and 95% CIs associated with each group are shown by the adjacent broken line. F. An estimation plot presents the preference index (PI) for each fly calculated by using the locomotion data from the whole epoch (45 seconds). The color-code, layout, and statistics are the same as Panel B.

**Author response image 3. sa2fig3:** Orco activation triggers attraction A-C-E. Trace plots representing the mean location of controls (*w^1118^; Orco-Gal4* and *w^1118^; UAS-CsChrimson*), and test flies (*Orco-Gal4 > UAS-CsChrimson*) throughout an experiment at 14, 42, and 70 μW/mm2 light intensities. The blue and orange ribbons indicate 95% CIs for the control and test flies, respectively. Second epochs were used to calculate the preference scores (shown in black rectangle). B. An estimation plot presents the individual preference (upper axes) and valence (lower) of flies with activatable Orco+ neurons in the WALISAR assay. In the upper panel, each dot indicates a preference score (wTSALE) for an individual fly: *w^1118^; Orco-Gal4* and *w^1118^; UAS-CsChrimson* flies are colored blue; and *Orco-Gal4 > UAS-CsChrimson* flies are in orange. The mean and 95% CIs associated with each group are shown by the adjacent broken line. In the bottom panel, the black dots indicate the mean difference (ΔwTSALE) between the relevant two groups: the valence effect size. The black whiskers span the 95% CIs, and the orange curve represents the distribution of the mean difference. D. A scatter plot shows how long it takes for each fly to encounter the optogenetic light once it is switched on. The median and 95% CIs associated with each group are shown by the adjacent broken line. F. An estimation plot presents the preference index (PI) for each fly calculated by using the locomotion data from the whole epoch (45 seconds). The color-code, layout, and statistics are the same as Panel B.

**Author response image 4. sa2fig4:** Or7a has neutral valence A-C-E. Trace plots representing the mean location of controls (*w^1118^; Or7a-Gal4* and *w^1118^; UAS-CsChrimson*), and test flies (*Or7a-Gal4 > UAS-CsChrimson*) throughout an experiment at 14, 42, and 70 μW/mm2 light intensities. The blue and orange ribbons indicate 95% CIs for the control and test flies, respectively. Second epochs were used to calculate the preference scores (shown in black rectangle). B. An estimation plot presents the individual preference (upper axes) and valence (lower) of flies with activatable Or7a+ neurons in the WALISAR assay. In the upper panel, each dot indicates a preference score (wTSALE) for an individual fly: *w^1118^; Or7a-Gal4* and *w^1118^; UAS-CsChrimson* flies are colored blue; and *Or7a-Gal4 > UAS-CsChrimson* flies are in orange. The mean and 95% CIs associated with each group are shown by the adjacent broken line. In the bottom panel, the black dots indicate the mean difference (ΔwTSALE) between the relevant two groups: the valence effect size. The black whiskers span the 95% CIs, and the orange curve represents the distribution of the mean difference. D. A scatter plot shows how long it takes for each fly to encounter the optogenetic light once it is switched on. The median and 95% CIs associated with each group are shown by the adjacent broken line. F. An estimation plot presents the preference index (PI) for each fly calculated by using the locomotion data from the whole epoch (45 seconds). The color-code, layout, and statistics are the same as Panel B.

**Author response image 5. sa2fig5:** Or69a+ neurons trigger no response A-C-E. Trace plots representing the mean location of controls (*w^1118^; Or69a-Gal4* and *w^1118^; UAS-CsChrimson*), and test flies (*Or69a-Gal4 > UAS-CsChrimson*) throughout an experiment at 14, 42, and 70 μW/mm2 light intensities. The blue and orange ribbons indicate 95% CIs for the control and test flies, respectively. Second epochs were used to calculate the preference scores (shown in black rectangle). B. An estimation plot presents the individual preference (upper axes) and valence (lower) of flies with activatable Or69a+ neurons in the WALISAR assay. In the upper panel, each dot indicates a preference score (wTSALE) for an individual fly: *w^1118^; Or69a-Gal4* and *w^1118^; UAS-CsChrimson* flies are colored blue; and *Or69a-Gal4 >UAS-CsChrimson* flies are in orange. The mean and 95% CIs associated with each group are shown by the adjacent broken line. In the bottom panel, the black dots indicate the mean difference (ΔwTSALE) between the relevant two groups: the valence effect size. The black whiskers span the 95% CIs, and the orange curve represents the distribution of the mean difference. D. A scatter plot shows how long it takes for each fly to encounter the optogenetic light once it is switched on. The median and 95% CIs associated with each group are shown by the adjacent broken line. F. An estimation plot presents the preference index (PI) for each fly calculated by using the locomotion data from the whole epoch (45 seconds). The color-code, layout, and statistics are the same as Panel B.

**Author response image 6. sa2fig6:** Or56a induces aversion response in flies A-C-E. Trace plots representing the mean location of controls (*w^1118^; Or56a-Gal4* and *w^1118^; UAS-CsChrimson*), and test flies (*Or56a-Gal4 > UAS-CsChrimson*) throughout an experiment at 14, 42, and 70 μW/mm2 light intensities. The blue and orange ribbons indicate 95% CIs for the control and test flies, respectively. Second epochs were used to calculate the preference scores (shown in black rectangle). B. An estimation plot presents the individual preference (upper axes) and valence (lower) of flies with activatable Or56a+ neurons in the WALISAR assay. In the upper panel, each dot indicates a preference score (wTSALE) for an individual fly: *w^1118^; Or56a-Gal4* and *w^1118^; UAS-CsChrimson* flies are colored blue; and *Or56a-Gal4 > UAS-CsChrimson* flies are in orange. The mean and 95% CIs associated with each group are shown by the adjacent broken line. In the bottom panel, the black dots indicate the mean difference (ΔwTSALE) between the relevant two groups: the valence effect size. The black whiskers span the 95% CIs, and the orange curve represents the distribution of the mean difference. D. A scatter plot shows how long it takes for each fly to encounter the optogenetic light once it is switched on. The median and 95% CIs associated with each group are shown by the adjacent broken line. F. An estimation plot presents the preference index (PI) for each fly calculated by using the locomotion data from the whole epoch (45 seconds). The color-code, layout, and statistics are the same as Panel B.

**Author response image 7. sa2fig7:** Flies avoid Or85d activation A-C-E. Trace plots representing the mean location of controls (*w^1118^; Or85d-Gal4* and *w^1118^; UAS-CsChrimson*), and test flies (*Or85d-Gal4 > UAS-CsChrimson*) throughout an experiment at 14, 42, and 70 μW/mm2 light intensities. The blue and orange ribbons indicate 95% CIs for the control and test flies, respectively. Second epochs were used to calculate the preference scores (shown in black rectangle). B. An estimation plot presents the individual preference (upper axes) and valence (lower) of flies with activatable Or85d+ neurons in the WALISAR assay. In the upper panel, each dot indicates a preference score (wTSALE) for an individual fly: *w^1118^; Or85d-Gal4* and *w^1118^; UAS-CsChrimson* flies are colored blue; and *Or85d-Gal4 > UAS-CsChrimson* flies are in orange. The mean and 95% CIs associated with each group are shown by the adjacent broken line. In the bottom panel, the black dots indicate the mean difference (ΔwTSALE) between the relevant two groups: the valence effect size. The black whiskers span the 95% CIs, and the orange curve represents the distribution of the mean difference. D. A scatter plot shows how long it takes for each fly to encounter the optogenetic light once it is switched on. The median and 95% CIs associated with each group are shown by the adjacent broken line. F. An estimation plot presents the preference index (PI) for each fly calculated by using the locomotion data from the whole epoch (45 seconds). The color-code, layout, and statistics are the same as Panel B.

3. ORN combinations: One of the key points of this manuscript is what it tells us about the rules by which ORN combinations work. While the authors show what their study rules out, we felt that they fall short of discussing what might be occurring instead. So, could the authors please include some discussion around this point?In this section we also recommend incorporating Sup Figure 8 into the main figure 5. It possible that different ORN pair use different interaction rules and the grouped analysis in figure 5 would mask this. Sup Fig8 is more informative in this regard.

We have incorporated the Supplementary Figure into Figure 5, and written a new Discussion section about possible mechanisms of valence computation.

4. Statistics: The statistics in this manuscript are quite involved. We recognise that the authors are promoting the use of Empirical-Bayes methods for statistical inferences. Since this is not commonplace, could the authors please incorporate an intuitive explanation about Empirical Bayes, its assumptions, and why it’s better suited to the analysis? We think this will greatly improve accessibility of this manuscript, and therefore its impact.

We thank the reviewing team for the suggestion. Since Empirical Bayes is a method with potential applicability for screens, including neurogenetic screens, we agree that it is important to explain the method in accessible terms.

We have added a version of the following explanation about Empirical Bayes, its assumptions, and why it’s better suited to the analysis.

“Empirical Bayes (EB) is a method for statistical inference using hierarchical Bayesian models. While relatively unknown in behavioral genetics, EB is widely used to filter omics data, having been originally developed for microarray data, and currently routinely used for, among other applications, mass spectrometry proteomics (Koh et al., 2019). This approach is intrinsically connected to the traditional analysis workflow based on hypothesis testing with multiple-testing correction, but with a major difference (Efron and Tibshirani, 2002). The hypothesis-testing approach is solely based on tail probabilities under the null hypothesis, and does not consider true signals. The key advantage of EB is that it explicitly models distributions of both the noises (null hypotheses), and the true signals (alternative hypotheses). Thus, for the analysis of a genetic screen, the key difference of this approach (compared to the conventional hypotheses testing-based analysis) is that we evaluate the significance of response levels across all ORNs simultaneously, abelled their distributions as a mixture of a random noise component and a real signal component. Empirically speaking, when the true effect sizes are modest, this typically increases sensitivity. Further, the mixture abelled naturally produces a confidence score, i.e. the posterior probability of true signal from the underlying Bayesian model, which is the inverse of the local false discovery rate (Efron, 2010).”

5. Comparison with the Bell and Wilson study: Could the authors please include the number of ORN pairs tested in the B and W study and their own (28 and 7)? With respect to the stimulation conditions listed in this table (Supp Table 4), we assume that the authors’ count of 6 conditions is because they are including with and without airflow for their 3 light intensities. In this case, the B and W study should be listed as 16. Alternatively (indeed preferably), the two studies should be listed as 8 and 3 respectively.

We thank the Reviewers and Editors for their guidance. The B and W study did an experiment with and without airflow only for *Orco-Gal4*, and then proceeded with airflow experiments from there onwards. The direct comparison is eight versus three light intensities. For the airflow, the comparison is 1 ORN (*Orco*) versus 45 ORNs.

We have added two rows and updated the numbers in Supplementary Table 4.

Reviewer #1 (Recommendations for the authors):Figure 1A I cannot tell the direction of airflow in the corridors. The air port is only shown on one side, does that mean airflow is unidirectional? Where is the exhaust?I would prefer a more schematic/conceptual drawing of the arena than this quasi-realistic one where the main feature that pops out to me are the flies themselves. I would prefer that the drawing conveys the technical details needed to evaluate what the flies experience in the assay.

An additional schematic was added to Figure 1.

Fig1F The y-axis label wTSALE should be swapped for a term with some intrinsic meaning. There isn’t even any basic description of what wTSALE means in the Results section, the reader has to go to the Methods. I think it would be helpful for the reader to understand the assay more clearly if the full description is in the Results. It is basically the proportion of time the flies spend outside the light, with the clock starting only after the fly has experienced the light the first time. That’s pretty easy to understand and the y-axis label could be % time outside light’ or even ‘preference light’ for positive values and ‘preference dark’ for negative values without criminally oversimplifying the measurement (IMO).

We thank the Reviewer for this helpful suggestion. We have improved the description of wTSALE to the Results, in a section titled “WALISAR data analysis: the wTSALE metric”.

Also I am not convinced the weighting the authors use (i.e. wTSALE vs TSALE) is really justified. Essentially they are trying to control for shorter sampling periods of the fly’s behavior in an extremely simple linear way. That implies that short sampling periods may not be representative – is it fair to simply weight those down so the score goes closer to zero since that actually indicates a lack of preference? Basically since any value on their y-axis carries meaning, it seems unfair to weight some points down simply because sampling wasn’t extensive enough. Why not just require some minimal time window for flies to have experienced the light (i.e. know what they are choosing) and look at the overall proportion of time in light vs dark?

Requiring a minimum time window and weighting the score are doing similar things, just that one is thresholded while the other is graded. The weighting was introduced as a non-thresholded way to allow all the available data to contribute in proportion to the time the animal was responding to the optogenetic light, which we consider to be the relevant metric. Regardless, as we show in the metric case study, the exact choice of metric does not dramatically alter the preference outcomes. We are aware that we made many assumptions and human decisions in the course of this project, however the use of wTSALE cannot account for inter-study differences.

Extended discussion: Extracting data from the final seconds of an experiment, and using this data to calculate a preference index is a common practice in the field. These brief arbitrary durations may range from as short as 5 s to 10 s or even just the final position (Aso et al., 2014; Dolan et al., 2019). While widely used, the conventional approach has potential issues, which encouraged us to implement wTSALE.

We investigated possible undersampling arising from using the last 3, 5, or 10 seconds of the optogenetic activation of Or67d neurons (Author response image 1). The overall conclusion remains the same across the three calculations in this example: that Or67d triggers attraction. This comparison shows that wTSALE is not masking or obscuring valence. However, there is a problem with the short-duration metrics: the majority of the flies have either a -1 or +1 preference index (PI). (Indeed, there are so many data points at the extremes, the swarm plot cannot display them). This distorted, tail-heavy, high-variability distribution shows that, for short-duration epochs, P.I. scores are heavily dependent on where in the chamber the flies happened to be for those few seconds (Author response image 1).

Instead of an under-sampled short epoch, one could use the entire test to calculate preference index. However this approach introduces a new pitfall: it would include all flies even if they never encountered optogenetic light—the intervention that is under investigation. As our analysis of the Or67d activation experiment demonstrates, when the whole epoch was used to measure preference, there are many flies with apparent “total aversion” (PI = –1; Author response image 1). Given that these flies never crossed into the illuminated side of the chamber and experienced activity in the Or67d neurons, it is misleading to conclude that these flies ‘avoid’ the activation of Or67d neurons 100% of the time.

The metric used in our study—wTSALE—utilizes all the locomotion data following a fly encountering the optogenetic light. It (1) eliminates the need for selecting an arbitrary cutoff; (2) uses all of the relevant tracking data for the analysis, thereby reducing sampling error and variation, while capturing the attraction response triggered by the Or67d neurons (Author response image 1).

We have shown the metrics case study here, which demonstrates that wTSALE is not misestimating valence when compared to more conventional metrics.

P.10 “This result indicates that wind has essentially no impact on ORN-elicited behavior in walking flies” this should read ‘single-ORN-elicited behavior’ since this is all the authors tested. ORN-elicited behavior could be read as ORN activation in general i.e. odor-based activation, where there is likely an effect of wind at least in some assays.

We thank the Reviewer for catching this. We have changed the text as suggested.

Figure 5: The authors analyze all ORN pairs together to test whether they summate/max pool/min pool but prior work (Bell and Wilson) showed that some pairs summate while others max pool, which would confound the style of grouped analysis in Figure 5G-I.

We agree with the Reviewer, which is why we also did the multiple linear regression.

Extended discussion: Of several analyses addressing this hypothesis, the multiple linear regression addresses this directly. The charts shown in Figure 5J display the axes and diagonal lines, abelled either “summation” or “max pooling”. If either/both of the simple rules governed ORN combinations, the β weights would be expected to cluster along these lines. Instead, we observe a broad scatter that shifts with increasing intensity. Both of these features (the locations of the weight markers and the fact that they shift) erode confidence in the either/or simple-rule hypothesis.

The present study was started a year before the publication of Bell and Wilson, and, while we realized early on that this study would inevitably be at least a partial replication, we never intended it to be a full-throated critique of the prior study. Bell and Wilson is an excellent, pioneering study. Nevertheless it is, like the present study, imperfect. We have not used effect sizes and other state-of-the-art analytic methods for their own sake, but because they have been established in the statistical literature to work better than *ad hoc* methods. To determine the combination type, Bell and Wilson used significance tests between the ORN-combination P.I. and two other metrics (the sum of the components and the larger component), combined with the calculation of *P* value ratios. This analysis shares all the well-documented problems associated with significant testing for confirmatory analyses, including false dichotomization. The primary B and W analysis workflow is: (1) continuous P.I. data is recast as binary outcomes (significant/not significant), and (2) this new binary data is then mapped to a dual mechanistic hypothesis (sum or max). However, this is based on the false assumption that significant differences are only compatible with the sum or max hypotheses.

It is likely that the B and W data, if it were analyzed with effect-size methods, would also show a variety of non-sum, non-max, weighted poolings. Indeed, two exemplar combinations (Bell and Wilson, Figure 5BC) have response magnitudes that exceed both models, and the overview matrix (Bell and Wilson, Figure 6A) shows that 11 of 28 tests were non-significant. We would have liked to re-analyze B and W’s data with best-practice methods, but when we requested the authors to share their data, they declined. Another barrier to a re-analysis is that their experiments included no Gal4 controls (not done) and very few UAS controls (Supplementary File 2 – Sheet 3), making it impossible to calculate effect sizes that correctly isolate the optogenetic influence.

We moved the former Figure 5GHI to the supplement, and Figure S8 to the main Figure.

Additionally since only 7 combinations were used and only 3 intensity levels, this figure is the weakest part of the paper, which up to this point has been extremely extensive. It also makes the first entry in Table 4 (Number ORNs tested = 45 vs n=8 for Bell and Wilson) unfair since Bell and Wilson actually looked at all combinations among 8 ORNs.

We have added a row to Supplementary File 2 – Sheet 3 describing the difference in combinations.

Can the authors discuss more about how the dominant β value can flip as the stimulation intensity increases? How would that work in terms of neural activity in the biological network? Also, what does the diversity of β values imply about biological network, does it potentially correspond to different weights on different downstream targets?

We thank the Reviewer for this valuable suggestion.

In the current experiment, the variability of β values across a range of stimuli indicates that we captured particular instances of ORN activation among many possible neural response mechanisms, and that many ORN combinations likely share downstream targets. Therefore we can cautiously infer that the underlying neural network is multifactorial and perhaps context-specific. However, we also believe that reverse-engineering the exact on-off combinations through estimated β values requires a wider set of experiments with distinct stimuli than we present. Attempting to translate the meaning of β shifts into the structure or variability of the neural network carries a high risk of over-interpretation.

We have added a section (“Possible models of ORN combination combination”) to mention possible mechanisms underlying the β shifts.

Ending on a negative result is a little disappointing – one positive point the authors could make is that (with one exception) the ORN combinations all transition abelle more max pooling at higher stimulation rates. This suggests an competitive interaction between channels, which is easy to imagine. Undoubtedly it is complex with different downstream targets having different rules, but this is one fairly consistent trend.

See above.

I should say that I found Bell and Wilson more convincing because they examine interactions for each ORN pair over a wider range of spike rates. Here there are just three points for comparison, and when my eyes look at Figure 5C-E it seems that there is not a lot of difference between the three interaction modes.

Although the interaction analysis in B and W does cover more light intensities, we consider the smoothness of these curves to arise from deprecated experimental and statistical practices. The curves use data from experiments that only indirectly refer to UAS controls (just N_flies_ = 88 total), and lack driver controls entirely. This is unusual for optogenetic experiments in *Drosophila,* which typically use either no-retinal controls (Badel et al., 2016) and/or driver and responder controls as was done in the present study.

The precision in the B and W response curves appears to be overrepresented in two ways. First, the optogenetic response should incorporate variance from both experimental and test animals; by omitting the controls (and therefore control variance) is expected to increase precision. Second, based on our reading of their Experimental Procedures, the optogenetic response in B and W appears to be averaged from the 16 technical replicates, potentially increasing apparent precision by √16, i.e. four-fold. With both statistical practices applied in tandem, it seems likely that the overall precision could be increased by as much as eight-fold.

Using published effects and our own preliminary control experiments as guidelines, the present study selected a sample size (N = 52, 52, 52 for test, UAS, and Gal4 control flies) that would be adequate to detect medium effect sizes. To our surprise, the effect sizes of the majority of ORNs were small and even undetectable using the sensitive EB method.

Finally, the authors should somehow incorporate FigS8 into the main text since I’m sure that the interaction mode depends on the pair of ORNs being examined.

We have incorporated Figure S8 in the main text.

Reviewer #2 (Recommendations for the authors):– I need to understand the raw data better. What are the flies actually doing here? In 1B, the example fly seems to be walking back and forth at about 0.1 Hz. Is this representative of the population? Do the flies ever not move at all? How is this outcome dealt with?

If the flies remain in the light (either moving or non-moving) they are counted as P.I. = 1.0. If they remain in the dark, and never encounter the light, they are excluded from analysis. Overall motionless flies were excluded from the analysis. While we share the Reviewer’s interest in odor-related locomotion, a detailed analysis of trajectories is beyond the scope of the present study. We agree that the data set is rich, and will make it freely available for others to download and analyze.

We have uploaded all tracking data to Zenodo (https://doi.org/10.5281/zenodo.3994033).

The methods mention that Empirical Bayes has a principled way of excluding outliers. What is that way?

The current implementation of the Ebprot software offers an option to exclude valence Z scores that are four reference standard deviations (RSD) away from the mean of the same ORN. Reference standard deviation is computed as the median of SDs in the Z scores across all ORNs

When the fly’s path enters into the illuminated region, it seems to immediately stop and walk back to the opposite wall, and then on its next two cycles it stops before entering the illuminated region. Is this because the light spreads or does the fly remember where it hit the light before?

This is an interesting question, and is beyond the scope of the present study. We are preparing several other studies that address this kind of question.

What happens when the illumination occurs when the fly is already on the illuminated side?

The fly is considered to have encountered the light, and the data for wTSALE is calculated from this.

The effect of Gr66a>Chr (1F) is much larger than any of the OR effects. What does a weaker avoidance response look like?

We present the Response plots here.

What does an attraction response look like? The mean +/- 95% CI plots of 1C-E do not answer these questions. More individual animal trajectories and population occupancy heat maps would help a lot. Exclusively compressing the data to the one wTSALE number may well be obscuring worthwhile features of the behavior. With a richer characterization of the behavior, it might be possible to reduce the sample size and simplify the statistics.

Compressing complex behavior to a P.I. metric has a long history across behavioral neuroscience. For example, the widely used *Drosophila* olfactory T-maze uses endpoint counts, without any tracking data ever collected.

– The statistical methods are unusual and seem unnecessarily complicated (at least to me). Further, why these were used instead of something more conventional? Readers (at least this reader) would benefit greatly from clear language giving an intuition for how Empirical Bayes works, what are its assumptions, and why it is superior to more conventional, easier-to-understand methods.

We thank the Reviewer for this helpful suggestion. We have clarified the EB method above, and added it to the Methods.

– The distribution of wTSALE in Figure 2 F and G is striking. In these plots, including the controls, there is a large mode at wTSALE=1. This mode is not apparent in the distributions of 1F. Why are control flies so much more attracted to light in these experiments?

It is true, these kinds of preference experiments often have heavy-tailed distributions, sometimes much more extreme than the examples in Figure 1F and as presented here. For this reason, we analyze the data with robust bootstrap methods. The δ curve shows, as per the central limit theorem, that the difference distributions are often approximately Gaussian despite these heavy tails.

How does Empirical Bayes deal with non-Gaussian distributions?

The Empirical Bayes inference method (implemented in Ebprot) estimates a mixture distribution with two components: the Gaussian null distribution and the non-Gaussian alternative distribution(s). In Figure 4C, for example, the middle part of the overall distribution corresponds to the Gaussian null distribution, whereas the red and green distributions are non-parametric alternative distributions. In sum, the overall distribution of valence Z-scores is therefore abelled as a flexible non-parametric distribution, not bound by Gaussian distribution assumption.

– What direction does the wind flow through the chamber? It appears to run perpendicular to the illumination axis. Could this matter? Does wind itself impact the locomotion of the flies? Since only δ-wTSALE is shown, it seems possible that wind may affect the behavior in a way that would obscure an effect. Here again it would be helpful to show more of what the flies are actually doing.

Wind runs through the narrow chamber from one end to the other, perpendicular to the light bands. We have not observed wind directly affecting locomotion. We are using the same wind speed as Bell and Wilson: 35 cm/s, around 0.7 knots.

– The authors invoke “complex circuit dynamics” to explain the results of the combined-receptor experiments. I’m not sure what the authors mean here. “Dynamics” implies that time-dependent processes determine valence. If this were the case, these experiments would show no effect, since the stimuli don’t recapitulate the dynamics of odor-evoked ORN activity. The discussion in a recent paper by Ron Yu’s group (Qiu et al., 2021; Current Biology), deals with the non-labelled-line-ness of the mouse olfactory system in a thoughtful way. A similar discussion would benefit this paper as well.

We thank the Reviewer for the suggestion. We added a Discussion section about possible circuit dynamics that support the observed results.

Reviewer #3 (Recommendations for the authors):1. The authors use a large array of GAL4 driver lines that they claim cover only the relevant ORN type. However, for most of these lines this was not examined. Although in the past such lines were used for behavior experiments, recent studies are much stricter with the use of driver lines. Many studies have demonstrated that even expression in a single neuron (other than the target neurons) either in the central brain or in the VNC can affect behavioral results. The authors therefore must show that the lines used in this study only label the target neurons either by providing adequate citations or by examining this directly with confocal stacks of both whole brain and VNC.

Ultimately, in every project, decisions must be made on how to allocate limited resources, and early on we decided to pursue a behavioral study, not an anatomical survey. Anatomical surveys of this scale are major undertakings (Couto et al., 2005; Fishilevich and Vosshall, 2005). While we appreciate that this is a potential concern, this is outside the scope of this project, and respectfully request that this is not made a prerequisite for publication.

We have added a description to the new Limitations section in the Discussion that highlights that non-antennal cells could be responsible for some of the effects observed.

2. The authors do not show the relevance of their optogenetic activation of ORNs to odor activation of ORNs. Previous studies have shown that optogenetic activation of ORNs generates a firing rate of approximately 30 Hz (Bell and Wilson, 2016; Fox and Nagel, 2021). In contrast, ORNs can reach firing frequencies of up to 250 Hz in response to odors (Hallem and Carlson, 2006). In addition, ORNs show temporal dynamics, whereas I presume that the continuous illumination generates a more uniform response. The authors briefly discuss this in the methods section. They claim that “continuous illumination is a more conservative method (Tumkaya et al., 2019)”. However, the same authors claim in their Tumkaya et al., 2019 manuscript that “These results suggest that neither stimulation type is necessarily superior to the other: static- or pulsed-light stimulation can capture more of the native responses than the other in inducing olfactory behavior, depending on the neuronal type”. The authors also claim that “We also benchmarked behavioral responses for the Orco neurons against results from a prior study that performed physiological recordings and used a different temporal structure (Bell and Wilson, 2016), finding that the WALISAR protocol has comparable sensitivity (Figure S3)”. The fact that both optogenetic activations has similar behavioral results does not imply any relevance to an olfactory cue.My main concern is that the current optogentic stimulation probably activates ORNs relatively weakly, thus mimicking low odor concentration. As low odor concentrations elicit in many cases only weak behavioral responses it is more than possible that the lack of behavioral effect is just due to “low concentration” and not an indication to the actual role of each ORN.Taken together, I think the authors should go the extra mileage and show some relevance to olfactory stimuli.

The reviewer is correct: neuronal photoactivation is not a naturalistic stimulus, but a physiological intervention. In this case, we are using a behavioral readout of the depolarizing input into the system. The use of simplified, well-controlled non-natural experimental preparations has a long tradition in the history of neuroscience. All such reductionist experiments have their limitations and serve as a counterpoint to more naturalistic experiments, which typically have the converse limitation of complexity.

Extended discussion: The relevance of the optogenetic screen to olfactory valence is supported by at least two lines of evidence.

First, of the ten ORN types identified by the screen, six ORN types have been previously identified by other groups (using odorants) as ORNs that drive attraction or aversion. The re-discovery of these ORN types lends further credibility to the idea that the optogenetic screen is relevant to olfactory preference. Considering the noise and poor reproducibility in behavioral experiments and neuroscience at large (Button et al., 2013), it is reasonable to characterize these independent replications as displaying remarkable fidelity.

Second, there is a point of reference with an excellent odor-based study that used an odorant-triggered, olfactory-receptor method of ORN activation: geosmin–Or56a heterologous olfactory stimulation (Chin et al., 2018). In comparison with this olfactogenetic assessment of ORN-evoked behaviors, there are five receptor types (35a, 42b, 47a, 67d, Gr21a/Gr63a) that are neutral in with the odorant-evoked intervention, but show valence in our assay (Supplementary File 1, Author Response Tables 1, 2). Of these, for the two receptor types covered by multiple studies (42b, Gr21/Gr63a), the olfactogenetic result is the outlier while our screen is compatible with the broad consensus (Supplementary File 1). These observations indicate that—when compared with a naturalistic mode of odorant activation—the Chrimson activation protocol used in the present study does not consistently underestimate ORN valences.

As experimental interventions, the complexity of naturalistic stimuli poses a challenge. For many odorants, receptor deconvolution is a complex problem that makes it challenging to unambiguously determine ORN→behavior causality. Many ORNs bind a multitude of odorants promiscuously. Even for particular odorant–ORN pairs with high affinity and specificity (abelled lines), it is difficult to completely rule out weak but nevertheless potentially widespread binding to ‘off-target’ ORNs.

We consider that the study already has relevance to olfactory processing.

We have added a passage to the new Limitations paragraph about the differences in firing rates between optogenetic and olfactory stimuli, and the important concern about relevance to olfactory stimuli. We request that odorant experiments are not made a requirement of publication.

**Author response table 1. sa2table1:** 

Receptor Valence Assay	Stimulus	Sex Stage	Reference			
Or7a	–	oviposition	olfactogenetics	F	adult	(Chin et al., 2018)
	o	two-choice	optogenetic	M	adult	* **this study** *
Or19a	o	oviposition	olfactogenetics	F	adult	(Chin et al., 2018)
	o	two-choice	optogenetic	M	adult	* **this study** *
Or22a	o	oviposition	olfactogenetics	F	adult	(Chin et al., 2018)
	o	two-choice	optogenetic	M	adult	* **this study** *
Or23a	o	oviposition	olfactogenetics	F	adult	(Chin et al., 2018)
	o	two-choice	optogenetic	M	adult	* **this study** *
Or35a	o	oviposition	olfactogenetics	F	adult	(Chin et al., 2018)
	+	two-choice	optogenetic	M	adult	* **this study** *
	o	oviposition	olfactogenetics	F	adult	(Chin et al., 2018)
Or42a	o	two-choice	olfactogenetics	F	adult	(Chin et al., 2018)
	o	two-choice	optogenetic	M	adult	* **this study** *
	o	oviposition	olfactogenetics	F	adult	(Chin et al., 2018)
Or42b	+	two-choice	optogenetic	M	adult	* **this study** *
Or43a	o	oviposition	olfactogenetics	F	adult	(Chin et al., 2018)
	o	two-choice	optogenetic	M	adult	* **this study** *
Or47a	o	oviposition	olfactogenetics	F	adult	(Chin et al., 2018)
	+	two-choice	optogenetic	M	adult	* **this study** *
Or47b	–	oviposition	olfactogenetics	F	adult	(Chin et al., 2018)
	+	two-choice	optogenetic	M	adult	* **this study** *
Or49a	–	oviposition	olfactogenetics	F	adult	(Chin et al., 2018)c
	o	two-choice	optogenetic	M	adult	* **this study** *
Or59c	–	oviposition	olfactogenetics	F	adult	(Chin et al., 2018)
	–	two-choice	optogenetic	M	adult	* **this study** *
Or65a	o	oviposition	olfactogenetics	F	adult	(Chin et al., 2018)
	o	two-choice	optogenetic	M	adult	* **this study** *
Or67b	–	oviposition	olfactogenetics	F	adult	(Chin et al., 2018)
	o	two-choice	optogenetic	M	adult	* **this study** *
Or67d	o	oviposition	olfactogenetics	F	adult	(Chin et al., 2018)
	+	two-choice	optogenetic	M	adult	* **this study** *
	–	oviposition	olfactogenetics	F	adult	(Chin et al., 2018)
Or71a	o	two-choice	olfactogenetics	F	adult	(Chin et al., 2018)
	o	two-choice	optogenetic	M	adult	* **this study** *
Or82a	–	oviposition	olfactogenetics	F	adult	(Chin et al., 2018)
	o	two-choice	optogenetic	M	adult	* **this study** *
Or83c	–	oviposition	olfactogenetics	F	adult	(Chin et al., 2018)
	+	two-choice	optogenetic	M	adult	* **this study** *
Or85a	–	oviposition	olfactogenetics	F	adult	(Chin et al., 2018)
	o	two-choice	olfactogenetics	F	adult	(Chin et al., 2018)
	o	two-choice	optogenetic	M	adult	* **this study** *
Or85d	–	oviposition	olfactogenetics	F	adult	(Chin et al., 2018)
	–	two-choice	optogenetic	M	adult	* **this study** *
Or88a	o	oviposition	olfactogenetics	F	adult	(Chin et al., 2018)
	o	two-choice	optogenetic	M	adult	* **this study** *
Or92a	o	oviposition	olfactogenetics	F	adult	(Chin et al., 2018)
	o	two-choice	optogenetic	M	adult	* **this study** *
Gr21a	o	oviposition	olfactogenetics	F	adult	(Chin et al., 2018)
/Gr63a	o	two-choice	olfactogenetics	F	adult	(Chin et al., 2018)
	–	two-choice	optogenetic	M	adult	* **this study** *

**Author response table 2. sa2table2:** Valence results in olfactogenetics and current studies.

	Olfacto Valent	Olfacto Neutral
Opto Valent	4 (2)	5
Opto Neutral	6	8

3. The authors own data raises potential problems with their approach. Some of the ORNs that are classified as driving aversion or attraction seem to change valence value they induce with the light intensity. For example, the authors report Or42b to drive attraction in agreement with published literature. However, at the strongest light intensity it is actually neutral. Similarly, the authors report Or85d to drive aversion. However, at the strongest light intensity it is also neutral. So, are these Ors “neutral”?

The data do not support these ORNs as neutral. The optogenetic data give robust signals in the EB screen, showing signed probabilities of +1.0 (42b) or –1.0 (85d) for some intensities. Both of these ORN-type results corroborate either the majority of published reports (42b) or the sole available published result (85d). Thus, the available data strongly support these two ORNs as being attractive and aversive, respectively.

4. The authors test a number of previously suggested linear models and find that they do not predict how two-ORN odor valence emerges from single-ORN valence. However, linear models were shown to be insufficient to predict odor valence (Badel et al., 2016). It is thus not surprising that these linear models failed.

We thank the Reviewer for their comment.

5. The authors use two databases, one of odor responses (Hallem and Carlson 2006) and one of behavioral responses (Knaden et al., 2012) along with a linear model to try and predict odor valence from ORN activity. However, as mentioned above linear models are not adequate for describing the relation between ORN activity and Odor valence. Furthermore, I think the Knaden et al., database is a wrong database to use in this context. Knaden et al., used a trap assay. In this assay, flies are captured in the trap after a single entrance to the odor source. Thus, exploratory behavior, in which flies examine the odor and then can decide to avoid it, cannot occur, and this assay is expected to be biased towards reporting odors as attractive. Indeed, this was the case in the Knaden et al., database in contrast to other published results. This database was suitable for the claim raised by Knaden et al., that looked only at the most aversive and attractive odors, but it cannot be used to try to predict any odor valence.

We respectfully disagree: we consider the title of Knaden *et al.,* (“Spatial Representation of Odorant Valence in an Insect Brain”) to be justified. A trap assay can be viewed as a valence assay that depends on movements between the two trap entry points, around which there are odorant gradients and/or intermingling plumes. Each fly interacts with and moves in these plumes. While each fly is associated with a single outcome, the proportion of outcomes over many flies is used to estimate a preference. Using count-type data is fairly common in fly behavior, such as the olfactory T-maze which uses the endpoint locations of flies as the end of a two minute epoch. As detailed above in the discussion around wTSALE, we prefer video analysis, but nevertheless consider that endpoint and count data are valid ways to measure valence.

6. The authors used cold anesthesia just prior to loading the flies to the chambers and only 30 second acclimation following the cold anesthesia. However, cold anesthesia is known to have effects on behavior, increasing response time, reducing locomotion and reducing overall responses (just a few examples, Barron, 2000; MacMillan et al., 2017, Trannoy et al., 2015). I think most studies today try to avoid cold anesthesia just before the experiment. My concern here is that the lack of effect for most ORNs, may arise from general behavior impairment. Can the authors give a few examples from the neutral ORNs without cold anesthesia?

Please see the above discussion about cold anesthesia.

7. The authors conclude that: 1. "the majority of primary olfactory sensory neurons have neutral behavioral effects individually". This conclusion (as mentioned above) is definitely correct for the optogenetic activation, but its relevance to odor valence is questionable. Furthermore, Badel et al., 2016 already demonstrated with actual odor stimuli that “We find that the behavior is accurately predicted by a model summing normalized glomerular responses, in which each glomerulus contributes a specific, small amount to odor preference.” Thus, the novelty of the current study is not large.

We thank the Reviewer for the summary of and their viewpoint on the excellent Badel *et al.,* study.

Their second conclusion is that "olfactory sensory neurons…participate in broad, odor-elicited ensembles with potent behavioral effects arising from complex interactions". I agree with them that olfactory coding is complex. However, they did not show any actual odor responses to support their claim, neither did they provided even one complex mechanism. I think that stating that olfaction is complex is just not enough.

Many possible neuronal mechanisms could generate the weighting shifts observed. We are reluctant to make over-reaching specific conclusions. We note that even in deep-learning neural nets that can perform impressive classification tasks—where there is complete knowledge of the ‘synaptic’ weights—there is typically little understanding how they do so (Knight, 2017; Zeiler and Fergus, 2014).

We have added a section to the Discussion that discusses possible mechanisms.

8. To my understanding the order of the β coefficients can affect the interpretation of the data. However, I could not find a reference for this in the methods. Can the authors please elaborate on this?

The question is ambiguous, answers to two possible interpretations:

– The ordering of the β weights in the regression does not affect the outcome.

– The ordering of the β weights in Figures 5 and S7 are the same.

References

Aso Y, Sitaraman D, Ichinose T, Kaun KR, Vogt K, Belliart-Guérin G, Plaçais P-Y, Robie AA, Yamagata N, Schnaitmann C, Rowell WJ, Johnston RM, Ngo T-TB, Chen N, Korff W, Nitabach MN, Heberlein U, Preat T, Branson KM, Tanimoto H, Rubin GM. 2014. Mushroom body output neurons encode valence and guide memory-based action selection in *Drosophila*. eLife 3:e04580.

Badel L, Ohta K, Tsuchimoto Y, Kazama H. 2016. Decoding of Context-Dependent Olfactory Behavior in *Drosophila*. Neuron 91:155–167.

Bernard C. 2019. Changing the Way We Report, Interpret, and Discuss Our Results to Rebuild Trust in Our Research. eNeuro. doi:10.1523/ENEURO.0259-19.2019

Button KS, Ioannidis JPA, Mokrysz C, Nosek BA, Flint J, Robinson ESJ, Munafò MR. 2013. Power failure: why small sample size undermines the reliability of neuroscience. Nat Rev Neurosci 14:365–376.

Chin SG, Maguire SE, Huoviala P, Jefferis GSXE, Potter CJ. 2018. Olfactory Neurons and Brain Centers Directing Oviposition Decisions in *Drosophila*. Cell Rep 24:1667–1678.

Claridge-Chang A, Assam PN. 2016. Estimation statistics should replace significance testing. Nat Methods 13:108–109.

Cleveland WS. 1994. The Elements of Graphing Data. AT&T Bell Laboratories.

Couto A, Alenius M, Dickson BJ. 2005. Molecular, anatomical, and functional organization of the *Drosophila* olfactory system. Curr Biol 15:1535–1547.

Cumming G, Calin-Jageman R. 2016. Introduction to the New Statistics: Estimation, Open Science, and Beyond. Routledge.

Dolan M-J, Frechter S, Bates AS, Dan C, Huoviala P, Roberts RJ, Schlegel P, Dhawan S, Tabano R, Dionne H, Christoforou C, Close K, Sutcliffe B, Giuliani B, Li F, Costa M, Ihrke G, Meissner GW, Bock DD, Aso Y, Rubin GM, Jefferis GS. 2019. Neurogenetic dissection of the *Drosophila* lateral horn reveals major outputs, diverse behavioural functions, and interactions with the mushroom body. eLife 8. doi:10.7554/eLife.43079

Editor. 2017. Show the dots in plots. Nature Biomedical Engineering 1:79.

Efron B. 2010. Large-Scale Inference: Empirical Bayes Methods for Estimation, Testing, and Prediction. Cambridge University Press.

Efron B, Tibshirani R. 2002. Empirical bayes methods and false discovery rates for microarrays. Genet Epidemiol 23:70–86.

Fishilevich E, Vosshall LB. 2005. Genetic and functional subdivision of the *Drosophila* antennal lobe. Curr Biol 15:1548–1553.

Gardner MJ, Altman DG. 1986. Confidence intervals rather than P values: estimation rather than hypothesis testing. Br Med J 292:746–750.

Ho J, Tumkaya T, Aryal S, Choi H, Claridge-Chang A. 2019. Moving beyond P values: data analysis with estimation graphics. Nat Methods. doi:10.1038/s41592-019-0470-3

Knight W. 2017. The Dark Secret at the Heart of AI. MIT Technology Review.

Koh HWL, Zhang Y, Vogel C, Choi H. 2019. EBprotV2: A Perseus Plugin for Differential Protein Abundance Analysis of Labeling-Based Quantitative Proteomics Data. J Proteome Res 18:748–752.

Root CM, Ko KI, Jafari A, Wang JW. 2011. Presynaptic facilitation by neuropeptide signaling mediates odor-driven food search. Cell 145:133–144.

Tukey JW. 1977. Exploratory Data Analysis. Addison-Wesley Publishing Company.

Zeiler MD, Fergus R. 2014. Visualizing and understanding convolutional networksComputer Vision – ECCV 2014, Lecture Notes in Computer Science. Cham: Springer International Publishing. pp. 818–833.